# Magnetothermal genetic deep brain stimulation of motor behaviors in awake, freely moving mice

Rahul Munshi[1†], Shahnaz M Qadri[1†], Qian Zhang[2], Idoia Castellanos Rubio[1], Pablo del Pino[3], Arnd Pralle[1*]

[1]Department of Physics, University at Buffalo, Buffalo, United States; [2]Department of Physics, Philipps University Marburg, Marburg, Germany; [3]CIC biomaGUNE, Paseo Miramón 182, San Sebastian, Spain

**Abstract** Establishing how neurocircuit activation causes particular behaviors requires modulating the activity of specific neurons. Here, we demonstrate that magnetothermal genetic stimulation provides tetherless deep brain activation sufficient to evoke motor behavior in awake mice. The approach uses alternating magnetic fields to heat superparamagnetic nanoparticles on the neuronal membrane. Neurons, heat-sensitized by expressing TRPV1 are activated with magnetic field application. Magnetothermal genetic stimulation in the motor cortex evoked ambulation, deep brain stimulation in the striatum caused rotation around the body-axis, and stimulation near the ridge between ventral and dorsal striatum caused freezing-of-gait. The duration of the behavior correlated tightly with field application. This approach provides genetically and spatially targetable, repeatable and temporally precise activation of deep-brain circuits without the need for surgical implantation of any device.
DOI: https://doi.org/10.7554/eLife.27069.001

**\*For correspondence:**
apralle@buffalo.edu

[†]These authors contributed equally to this work

**Competing interests:** The authors declare that no competing interests exist.

## Introduction

Identifying neural circuitry controlling specific behaviors requires the ability to communicate with specific neurons in the brains of awake, freely moving animals. The most established interface is electrodes implanted in the brain. They provide rapid neuronal stimulation and recording, however, at the cost of being invasive and with limited spatial and cell type targeting. Optogenetics, visible light-based stimulation of neurons genetically engineered to express light-gated channels and ion pumps, provides genetic targeting of specific cell types in combination with rapid stimulation (*Banghart et al., 2004*; *Boyden et al., 2005*). The method quickly gained popularity and has been used in various animal models to study brain circuits associated with Parkinson's Disease (*Gradinaru et al., 2009*; *Kravitz et al., 2010*), addictive behavior (*Chen et al., 2013*; *Stefanik et al., 2013*), depression (*Nieh et al., 2013*; *Zhu et al., 2016*) and anxiety (*Tye et al., 2011*). However, optogenetics requires permanent brain implants to guide the light into the brain, and typically, an optical fiber, tethering the animal to the light source. A third approach, pharmacological neuromodulation using Designer GPCRs Exclusively Activated by Designer Drugs (DREADDs), provides a tetherless alternative to optogenetics, but generates a much slower response, ranging from tens of minutes to hours, as the drug must diffuse to the location of action (*Armbruster et al., 2007*; *Urban and Roth, 2015*). The ideal experimental approach to deep brain stimulation would combine minimal invasiveness - to avoid perturbing the behavior, sufficient speed - to permit real-time response, and cell-specific targeting.

Magnetic-field-based stimulation has considerable potential to provide fast, tetherless, and implant-free deep brain stimulation because magnetic fields are minimally scattered, minimally

absorbed by tissue, and travel freely through space (*Walsh and Cowey, 2000*). These characteristics of magnetothermal stimulation simplifies a range of experimental assays, which are challenging to perform using current techniques, and permit new types of experiments: The absence of tethers will permit novel social behavioral assays in a group of animals freely interacting in an arena, while only a subgroup of them gets stimulated. As magnetic stimulation only occurs when the animal resides within the magnetic field, the field geometry can be adapted to coincide with another signal for conditional place preference or feeding assays. Similarly, magnetically stimulating multiple sites in one brain to study network connectivity requires only multiple injections of solutions, possibly along a shared injection path, as opposed to multiple cannula implantations. Furthermore, the absence of any physical connection from brain to skull minimizes tissue damage and immune response (*Chen et al., 2015*). Lastly, assays using animals easily startled by the flashes of visible light used in optogenetics, such as Zebrafish, would benefit from magnetic stimulation. Therefore, a reliable magnetic field stimulation method would not only simplify experiments, currently performed using existing techniques, but enable a series of novel assays.

The challenge in implementing magnetic stimulation has been to develop a transducer capable of harnessing the energy of the magnetic field and translating it into a robust biological signal with high temporal and spatial specificity. Magnetic fields only interact with magnetic dipoles. The overall magnetic dipole of a particle is caused by the particle's intrinsic magnetic moment, which results from the orientation of a large number of spins of the ions making up the nanoparticle. The spins in a ferromagnet interact strongly, causing their parallel alignment to persist over time, and creating a permanent magnetic moment. In ferromagnetic particles, the magnetic moment is fixed to the particle crystal structure, causing a net magnetic dipole of the particle, which interacts strongly with external magnetic fields. However, the permanent dipoles of individual particles interact also in absence of an external field, causing aggregation of particles, rendering them unsuitable for neurostimulation.

In very small particles made from materials that are ferromagnetic in bulk, the spins interact, but at room temperature the direction of their magnetic moment relative to the particle fluctuates rapidly (*Brown, 1963*). After an external magnetic field aligns the moments, this alignment persists for a brief moment of time to permit the interaction. Such particles are called superparamagnetic (*Bean and Livingston, 1959*). They are the preferred transducers for magnetic biostimulation (*Xu and Sun, 2013*; *Pankhurst et al., 2009*). Superparamagnetic particles can been used as a transducer of magnetic field energy exploiting two different mechanisms: In a strong external DC field, the induced magnetic dipoles in neighboring NPs interact sufficiently strongly to aggregate the NPs and a field gradient exerts a small force (*Mannix et al., 2008*). Alternatively, our group showed that applying an alternating magnetic field (AMF) to superparamagnetic nanoparticles may be used to locally generate heat, which then actives nearby temperature-sensitive TRPV1 channels (*Huang et al., 2010*). This later approach, when used to stimulate TRPV1$^+$ neurons, is referred to as 'magnetothermal genetics' (*Chen et al., 2015*; *Huang et al., 2010*; *Stanley et al., 2012*).

Activating a channel requires overcoming an energy barrier slightly larger than the thermal fluctuation energy ($E_{thermal} = k_B T \sim 4 \cdot 10^{-21} J$ at 37°C, with Boltzmann constant $k_B$). The maximally delivered energy per transducer particle depends on the mechanism of interaction with the magnetic field. In the case of dipole-dipole interaction of two NPs with magnetic dipole $m$ and a distance $r$ apart, their interaction energy is $E_{diople} \sim \mu_0 m^2 / 4\pi r^3$ ($\mu_0$ being the vacuum permeability). In a 0.1 T external field, the magnetic dipole of two core-shell NPs used in this study is strong enough that their $E_{dipole}$ is larger than the thermal energy as soon as they are closer than 22 nm (*García-Prieto et al., 2016*; *Zhang et al., 2015a*) (*Figure 1—figure supplement 1A B*). However, for horse ferritin the magnetic dipole at 0.1 T per particle is so weak that the dipole interaction energy is a lot smaller than the thermal energy even if the ferritin molecules were touching each other (*García-Prieto et al., 2016*; *Zhang et al., 2015a*) (*Figure 1—figure supplement 1B*).

In magnetothermal stimulation, the transducer converts the energy of the external alternating magnetic field into heat. This process depends on the frequency $f$ and field strength $H$ of the external magnetic field, the time $\tau$ over which thermal fluctuations randomize the orientation of the moments in the particle, the particle volume and magnetization. The synthesized nanoparticles used in this study, heat by 450 W/g at 500 kHz and 15 kA/m, while horse ferritin does not heat in the same AMF (*Figure 1—figure supplement 1C*). Even if increasing the frequency driving ferritin to 50

MHz and 0.15 kA/m horse ferritin heats less than 1 W/g. In this comparison, the product $H \cdot f$ is left constant, as it is a measure for background effects in tissue (*Atkinson et al., 1984*; *Chen et al., 2015*). Hence, transducers coupling either via dipole-dipole or magnetothermal mechanism, deliver sufficient energy if using consisting of synthesized superparamagnetic NPs. This is in contrast to the unclear physics of ferritin-based magnetogenetics (*Meister, 2016*; *Stanley et al., 2015*; *2012*; *Wheeler et al., 2016*).

In magnetothermal genetic stimulation, the heat needs to be delivered to the TRPV1 channel efficiently, avoiding excess heat loss to the surrounding buffer fluid. Classically, magnetic nanoparticle-based hyperthermia heating uses a dense suspension of nanoparticles, a ferrofluid, in which the neighboring MNPs screen the cooling. An isolated MNP delivers only a few femtowatts of heat, which dissipates rapidly into the volume of water. In a two-dimensional dense array of MNPs on the membrane, the neighboring MNPs screen the lateral cooling, leaving only the direction normal to the membrane for heat dissipation. Membrane-bound MNPs deliver sufficient power to raise the temperature in a thin sheet along the membrane (*Huang et al., 2010*). Therefore, we aimed to use AMF heating of membrane-targeted MNP to evoke behavior in awake animals.

Here, we report the first robust and repeated magnetothermal genetic activation of motor behavior in awake, freely moving mice using magnetic nanoparticles (MNPs), attached to the plasma membrane of temperature-sensitized TRPV1[+] neurons, deep inside the brain. We demonstrate successful magnetothermal activation of three separate brain regions, motor cortex, dorsal striatum and the ridge between dorsal and ventral striatum. All three have been previously activated either optogenetically (*Gradinaru et al., 2007*; *Kravitz et al., 2010*) or chemogenetically (*Arenkiel et al., 2008*). We observed the same behavioral responses as had been reported resulting from optogenetic or chemogenetic neurostimulation in those areas. We were able to achieve successful neurostimulation in vivo without raising the overall tissue temperature by binding MNPs to the neuronal membrane. This magnetothermal method does not require any fixed implants or tethers, thereby permitting repeatable stimulation with real-time response monitoring and recording in awake freely moving mice capabilities.

## Results

### Principles of magnetothermal genetic neurostimulation

To achieve fast and robust neuronal stimulation in awake, freely behaving mice, we depolarized heat-sensitized (TRPV1[+]) neurons via magnetic-field-induced heating of superparamagnetic MNPs (*Figure 1A,B,F*). The MNPs were targeted to the neuronal membrane via an A2B5 antibody specific to neuronal glycosylated membrane proteins. This approach provided specific and dense labeling of neuronal membranes (*Figure 1E*), minimizing the amount of MNPs required for activation.

### Development and synthesis of an optimal magnetothermal transducer

An effective magnetothermal transducer must convert the alternating magnetic field energy efficiently into heat, be targetable to specific cells and sufficiently small to easily diffuse between the neurons in the brain (<30 nm) (*Thorne and Nicholson, 2006*). Exchange-coupled core-shell MNPs combining soft and hard magnetic materials (Cobalt Oxide and Manganese oxide) in a core-shell geometry, permit independent optimization of the MNP's size and magnetic properties (*Lee et al., 2011*; *Zhang et al., 2015b*). Thus, these MNPs can be tuned to heat efficiently at magnetic field frequencies, known to safely penetrate tissue (<1 MHz) (*Young et al., 1980*), while maintaining a diameter of less than 30 nm.

We synthesized MNPs with an 8.0 ± 1.0 nm Co-ferrite core surrounded by a 2.25 nm Mn-ferrite shell. To confer colloidal stability in physiological buffers, these MNPs were then encapsulated in the polymer PMA (dodecyl-*grafted*-poly-(isobutylene-*alt*-maleic-anhydride) (*Lin et al., 2008*). The PMA coating added 5.4 ± 1.4 nm to the MNP diameter (*Figure 1D*). We carefully analyzed the experimental magnetic and heating properties of these MNPs and found the results to be well described by the theory of superparamagnetic NPs (*Zhang et al., 2015b*). The efficiency of particular MNPs in converting AMF power to heat is measured and quantified as specific loss in power (SLP) in W/g. The MNPs used in this study have an SLP of 733.3 ± 2.8 W/g at 37.0 kA/m and 412.5 kHz, close to the theoretical maximum for particles of this size (*Zhang et al., 2015a*).

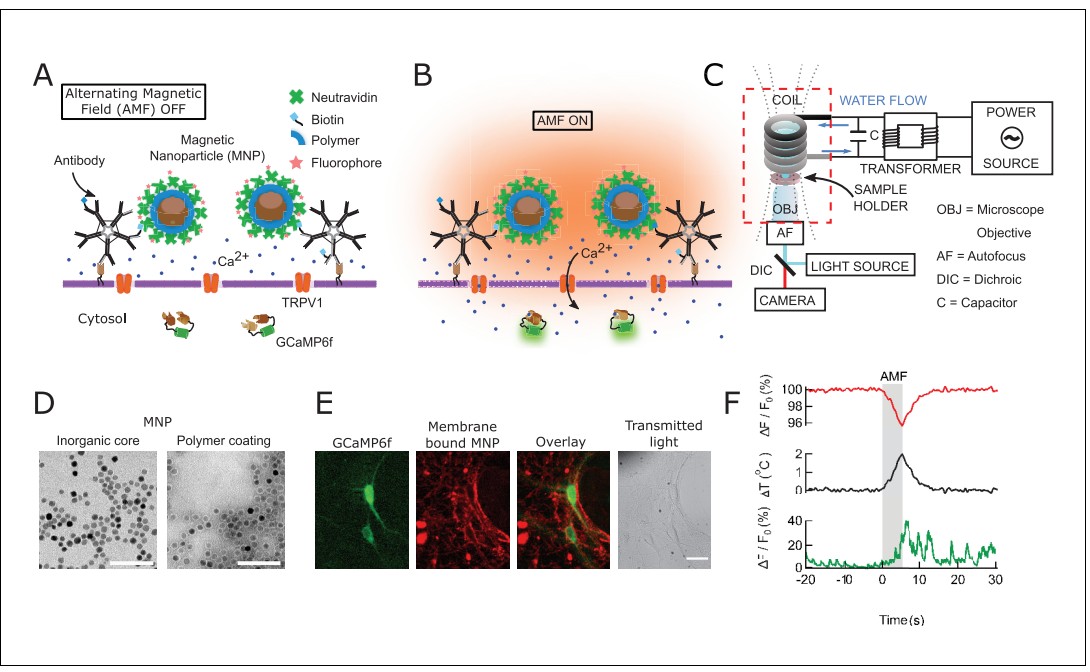

**Figure 1.** Magnetothermal genetic neurostimulation activates TPRV1 channels by heating membrane-bound magnetic nanoparticles using an alternating magnetic field. (**A**) Magnetic nanoparticles (MNPs) (brown), encapsulated in PMA polymer (blue ring) are functionalized with NeutrAvidin (green), conjugated with Dylight550 fluorophores (red stars), and attached to the neuronal membrane via biotinylated antibodies targeting membrane proteins. The neurons are transfected to express temperature-sensitive TRVP1 channels and the calcium indicator GCaMP6f. (**B**) Applying an alternating magnetic field ('AMF on') heats the membrane-bound MNPs. This heat dissipates, raising the temperature locally near the membrane, which activates the TRPV1 channels. The resulting calcium influx depolarizes the neurons and is measured as a transient intensity increase of the GCaMP6f fluorescence. (**C**) The experimental setup combining the alternating magnetic field (AMF) application with fluorescence microscopy for in-vitro studies. The AMF (dotted lines) is produced by a five turn, continuously water cooled coil made of copper pipe. The coil and capacitor C form an electrical resonator that is driven by a 7.5 kW alternating power source. Neurons grown on cover glass are placed directly underneath the coil in a non-metallic sample holder. The AMF causes eddy currents in metal parts, including the microscope objective (OBJ). Any focus drifts are compensated by a fast laser autofocus (AF) (also see *Figure 1—figure supplement 4*). Components within the red, dashed box are to scale. (**D**) Transmission electron micrographs showing 12.5 ± 1.2 nm core-shell MNPs. (Left) MNPs as synthesized. (Right) Negative staining visualizes the PMA polymer shell encapsulating the dark inorganic nanoparticles. Scale bar is 100 nm long. (**E**) From left to right: fluorescence micrographs of GCaMP6f[+] (green) neuron; labeled with MNPs (red); overlay of the GCaMP6f (green) and MNP (red) signals; and transmitted light image of the same neurons. Scale bar 10 µm (See also *Figure 1—figure supplement 2A*). (**F**) (Top) Local heating of MNPs during AMF application measured as a dip in DyLight 550 fluorescence intensity (red trace), which drops linearly with increasing temperature. The grey bar indicates the application of the AMF (22.4 kA/m, 412.5 kHz). (Middle) Temperature change near MNPs, as calculated from the fluorescence data using the calibration shown in *Figure 1—figure supplement 3* (black trace). (Bottom) The GCaMP6f fluorescence signal recorded in the neuron decorated with nanoparticles shows a Calcium transient after 5 s of AMF when the membrane temperature increased by 2°C. Temperature decreased after the AMF was removed and the Calcium transients slowly subsided again.

DOI: https://doi.org/10.7554/eLife.27069.002

The following figure supplements are available for figure 1:

**Figure supplement 1.** Magnetic properties of nanoparticles and ferritin.
DOI: https://doi.org/10.7554/eLife.27069.003
**Figure supplement 2.** Control experiments in HEK293T cells.
DOI: https://doi.org/10.7554/eLife.27069.004
**Figure supplement 3.** Calibration for in-situ temperature measurements.
DOI: https://doi.org/10.7554/eLife.27069.005
**Figure supplement 4.** Imaging set-up compatible with AMF heating.
DOI: https://doi.org/10.7554/eLife.27069.006

To target the MNPs specifically to neurons, we functionalized them with NeutrAvidin which then binds to biotinylated antibodies. We used either antibodies against endogenous neuronal surface markers, such as anti-CD56 (Thy-1), anti-CD90 (NCAM) or anti-A2B5, or against a fluorescent protein tag. The later allowed genetic targeting of specific cells, transiently expressing a surface protein with an extracellular GFP or mCherry (*Figure 1—figure supplement 2A and B*).

The local temperature rise in the vicinity of the nanoparticles was measured by fluorescence intensity changes of a Dylight 550 fluorophore integrated in the NeutrAvidin coating. The fluorescence intensities of molecular fluorophores are sufficiently temperature dependent to provide a molecular scale thermometer (*Huang et al., 2010*). We found that the fluorescence intensity of DyLight 550 decreased by 2.2% per degree temperature rise (*Figure 1—figure supplement 3*).

## Membrane-bound nanoparticles produced efficient, well localized heating

To open a sufficient number of TPRV1 channels to affect neuronal firing, the cell surface temperature needs to rise only a few degrees above the physiological temperature (*Huang et al., 2010*; *Chen et al., 2015*). First, we determined the efficacy of AMF-driven heating of MNPs in suspension, aka ferrofluid, and then separately of membrane-bound MNPs, using identical experimental conditions and AMF (*Figure 2*). Bath temperature was measured continuously using an immersed fiber optic thermometer (*Figure 2A*), while the membrane temperature was measured using the fluorescence intensity of DyLight 550 integrated in the NeutrAvidin coating of the membrane-bound MNPs (*Figure 2B*). The AMF (22.4 kA/m at 412.5 kHz) was generated in a custom built, water-cooled coil, surrounding the sample dish, which was mounted on an inverted microscope.

In the MNP suspension sample, the entire bath temperature (along with the cells) rose by two degrees (*Figure 2C*). Following a seven second AMF heating, the MNP suspension required more than 1 min to cool back to the initial temperature. Conversely, when the MNPs were bound to the cell membrane, only the membrane, and not the bath, was heated by several degrees (*Figure 2D*). After removing the AMF, the membrane temperature returned to the initial value within a few seconds. Hence, using membrane-bound MNPs provides a temporal precision of few seconds for terminating the magnetothermal neurostimulation.

The rates of temperature increase depend on the concentration of MNPs. For various volume concentrations of MNPs we measured rates from 0.1 to 0.5°C/s (*Figure 2E*). For membrane-bound MNPs, the rates were 0.1 to 1.0°C/s, depending on the area density of MNP (*Figure 2F*). Plotting the heating rate as a function of MNP concentration collapses these measurements. For suspension heating, the heating rate depends linearly on the MNP concentration and is fit to $5.8 \times 10^{-4} \pm 1.9 \times 10^{-6}$°C.$\mu m^3$/s (*Figure 2G*). Hence, the heating power per MNP is 2.55 ± 0.94 fW/particle, consistent with the prediction from magnetic measurements, 3.34 fW/particle (*Zhang et al., 2015a*). In the membrane-bound case, the graph of the heating rate versus MNP area density is fit to $1.1 \times 10^{-3} \pm 6.7 \times 10^{-5}$ C.$\mu m^2$/s (*Figure 2H*). Comparing the fit obtained for the membrane heating to the suspension heating shows that the volume of water heated in the membrane case corresponds to a 0.5-µm thick slab of MNP suspension. In reality, the temperature away from the membrane drops inversely with the distance from the membrane (*Baffou et al., 2013*). *Figure 2I and J* visualize MNP spacing for membrane and suspension cases, assuming homogenous MNP distribution.

To determine a lower bound for the field strength of AMF required for local membrane heating, we compared the membrane temperature increase in HEK cells decorated with MNPs at two different field strength, 12 kA/m and 30 kA/m (both at 412 kHz). A 30 s application of the weaker AMF heated the membrane as much as a 5 s application of the stronger AMF (*Figure 2—figure supplement 1*). However, at 12 kA/m AMF, once the local temperature has rising more than 2°C above the environment, cooling to the environment becomes comparable to the heating. Hence, even during longer AMF applications the maximal local temperature increase will remain in a safe range for cells.

Overall, we found that for membrane-bound MNP, the duration of AMF to achieve the two-degree rise necessary to activate TPRV1 channels varies between 2.1 s and 4.2 s (*Figure 2F*). The amount of MNPs required for similar heating rates in suspension heating is a few orders higher, 20–30 mg/ml (*Figure 2E* – H).

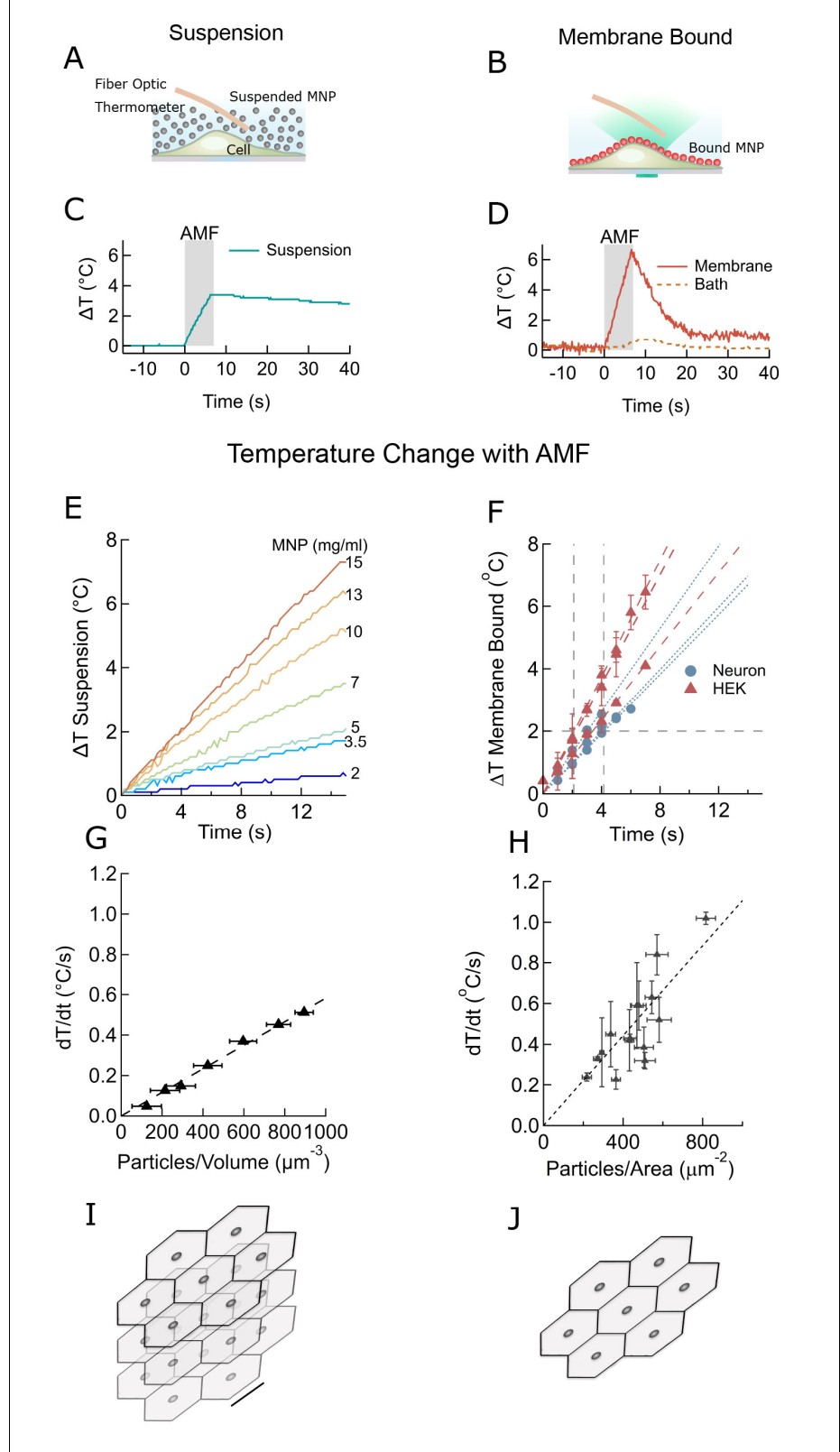

**Figure 2.** Membrane-bound MNPs heat the membrane locally without bath heating, cool quickly to turn signal off, and can heat more efficiently than MNP suspensions. (**A**) Illustration showing MNP suspension in the bath above cells. During the AMF application the MNPs heated, raising the temperature of the entire bath volume, as measured by the fiber optic thermometer. The entire left column of this figure focuses on heating of a MNP

*Figure 2 continued on next page*

*Figure 2 continued*

suspension, while the right column compares this to heating the same MNPs when membrane bound. Not to scale. (**B**) Illustration showing MNPs bound via antibodies to the cell membrane. MNPs form a two-dimensional sheet along the contour of the cell membrane. AMF nanoparticle heating is limited only to the immediate vicinity of the cell membrane. DyLight fluorophores were attached to the MNPs (orange MNPs) to measure the local temperature changes near the membrane-bound MNPs via fluorescence microscopy. The bath temperature was monitored using the optic fiber thermometer. Not to scale. (**C**) Temperature rise and subsequent cooling during a 7 s AMF (22.4 kA/m at 412.5 kHz) application in a MNP suspension (10 mg/ml) above the cells (blue). (**D**) Temperature rise and subsequent cooling during a 7 s AMF (22.4 kA/m at 412.5 kHz) application on cell-membrane-bound MNPs (orange solid, recorded via fluorescence). Simultaneous recording of the bath temperature ((no perfusion, orange dashed line) confirmed that the MNP heating remains confined near the cell membrane. This is unlike the suspension heating of MNPs, where the entire bath heats uniformly. Another contrasting feature of membrane-bound MNP heating seen here is the rapid cooling. (**E**) Temperature rise in suspensions of $CoFe_2O_4 – MnFe_2O_4$ core-shell MNPs at various concentrations plotted versus time. Numbers to the right indicate MNP concentration in mg/ml (AMF in all cases 22.4 kA/m at 412.5 kHz). (**F**) Heating rates of bound MNPs are dependent on the area density on the membrane. Temperatures recorded at various time points are linearly fitted (dashed lines). Blue markers indicate data collected from neurons, using biotinylated anti-A2B5 for particle binding. Red markers indicate data collected from HEK293T cells with MNP attached to enzymatically biotinylated surface proteins. Error bars incorporate measurement errors and error associated with temperature calibration from fluorescence measurement. (see also *Figure 1C*). (**G**) Rate of temperature rise plotted as a function of density of MNPs in suspension. Error bars indicate error in estimation of concentration. Y errors are smaller than the marker size. Dashed line shows the linear fit. (**H**) Temperature rise around MNPs bound to cell membranes plotted against the mean area density of the MNP distribution. X error bars indicate the error in estimation of MNP density and Y errors are obtained as mentioned in (**F**). (**I**) Stacked sheets of isotropically distributed nanoparticles. Scale bar is 100 nm long. The nanoparticle density shown, corresponds to that required for 0.5 °C/s rise in suspension temperature. (**J**) Individual sheets from (**I**). Same scale. The density is enough to heat the membrane by 0.5 °C/s.

DOI: https://doi.org/10.7554/eLife.27069.007

The following figure supplement is available for figure 2:

**Figure supplement 1.** Effect of AMF strength on local heating.
DOI: https://doi.org/10.7554/eLife.27069.008

## Thermal activation of TPRV1⁺ hippocampal neurons

To measure the response of TRPV1[+] hippocampal neurons to thermal stimulation, we used the fast genetically encoded calcium indicator GCaMP6f (*Chen et al., 2015*; *Grewe et al., 2010*), and deconvolved the $Ca^{2+}$ transients to uncover the underlying spike train (*Figure 3—figure supplement 1*). First, we investigated the spontaneous firing rate of 10 day old TRPV1[+] hippocampal neurons across a range of physiologically relevant bath temperatures. From 32°C to 36°C, the Calcium spiking rate was temperature independent. As bath temperature was increased from 37°C to 39°C, spontaneous activity of TRPV1[+] neurons increased from $1.5 \pm 1.2$ action potentials (APs) per 5 s to $13 \pm 3$ APs in the same period. At 39°C, only about 15–20% of rat TRPV1 channels open (*Grandl et al., 2010*), but the resulting Calcium influx is sufficient to trigger neuronal activity because TRPV1 has an about 1,000-fold higher single channel conductance than ChR2 (*Lórenz-Fonfría and Heberle, 2014*; *Studer and McNaughton, 2010*). In control neurons, without TRPV1, we observed a slight decrease of the spiking rate, to $1 \pm 1$ APs per 5 s (*Figure 3A*). Detailed analysis of single calcium transients in the absence of TRPV1, and with TRPV1 at channel activating temperatures showed, that, within the resolution of GCaMP6f-based calcium imaging, TRPV1 activity does not change the shape of the calcium transient and the underlying AP (*Figure 3—figure supplement 1*). Therefore, expressing of TRPV1 in hippocampal neurons renders their firing rate highly heat-sensitive without disrupting natural function.

## Membrane-localized MNP heating stimulates cultured hippocampal neurons

To measure the MNP heating evoked stimulation, we transfected cultured hippocampal neurons with TRPV1 and GCaMP6f, then decorated the plasma membrane with MNPs. The red fluorescence of the DyLight550-labeled MNPs was detected all along the neuronal membrane, indicating good surface labeling (*Figure 1E*).

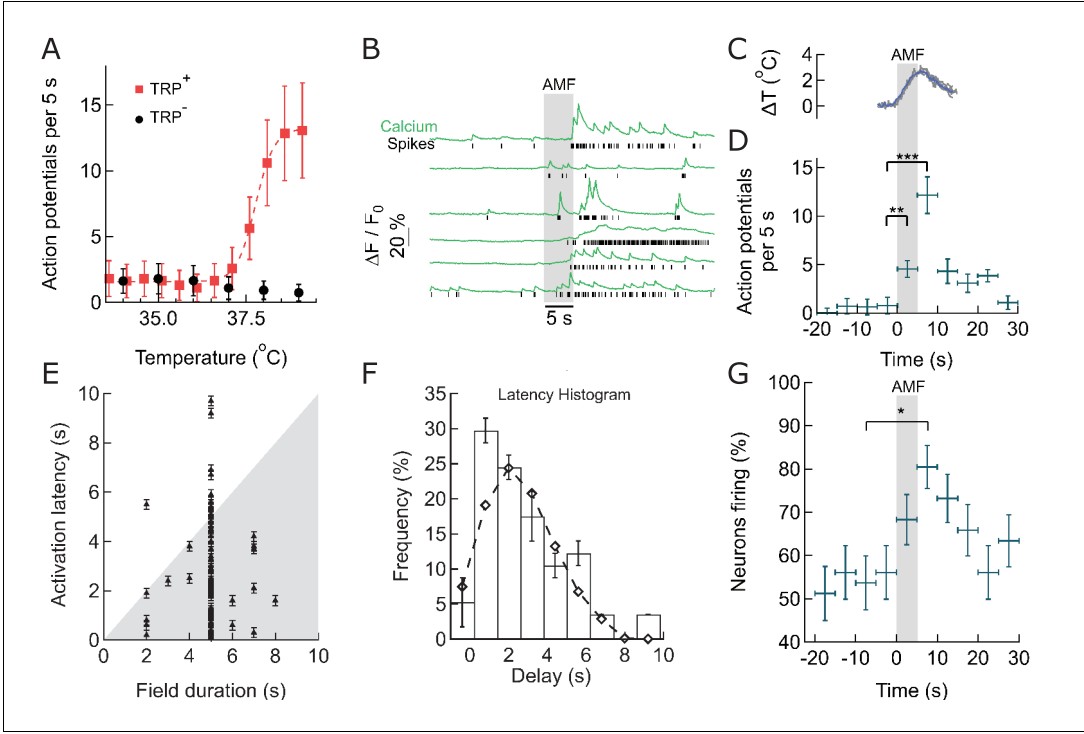

**Figure 3.** Within seconds of AMF application, membrane-targeted MNP stimulate magnetothermally TPRV1+ neurons in culture. (**A**) Rate of Action Potential firing as a function of bath temperature, recorded from GCaMP6f transients observed in TRPV1 expressing hippocampal neurons (red) and wild-type neurons (control, black) when perfused with pre-heated buffer. The Ca²⁺ transients are modeled by a spike train (see ***Figure 3—figure supplement 1***). The data points are fitted with Hill equation, giving a midpoint of 37.7 ± 0.06°C, which corresponds to half maximal firing rate (dashed curve). (**B**) GCaMP6f fluorescence intensity changes (green, Ca²⁺) in different TPRV1 +neurons decorated with MNP (5 s field, 22.4 kA/m at 412.5 kHz, in gray). Calculated spike events (black) are indicated under each Ca²⁺ trace (see ***Figure 3—figure supplement 2***). (**C**) Change of cell surface temperature as measured by DyLight550 fluorescence (average of three experiments). (**D**) GCaMP6f signal recorded from nanoparticle-coated TRPV1 +neurons binned in 5 s intervals, indicated by x error bar (mean ±s.e.m, 13 neurons). The spiking frequency increased from 1.8 ± 0.6 per 5 s, before AMF (field 0 to 5 s), to 4.5 ± 1.2 per 5 s, during the AMF (n = 13, \*\*p=0.0028, unpaired T-test), and 12.1 ± 2.0 per 5 s immediately following the AMF (n = 13, \*\*\*p=0.0002, unpaired T-test, 95% confidence intervals [1.5,2.23] and [10.8,13.3]) (Supplement 2). (**E**) Plot of activation latency, time interval between onset of field stimulation and first AP detect versus field duration. All data points lying on the gray background indicate that the first spike was detected while the field was still on (87% of events, n = 79, six cultures). Error bars indicate the temporal measurement uncertainty. (**F**) Percentage of neurons firing their first AP after field onset in the time interval indicated (subset of 41 cells from A which were stimulated for 5 s). The histogram was fitted (no weighting) with a Poisson curve (λ = 2.18 ± 0.17 s). Error bars are obtained as difference in population between bins shifted to left and right, following temporal uncertainty as indicated in (**E**). (**G**) Percentage of active neurons in each time interval (Alternating magnetic field applied 0–5 s). Error bars: x indicate 5 s time bin, y as in (**F**). A five-second AMF application increased the active population from 53.7 ±1.6% to 80.5 ±5.1% of neurons, \*p=0.032, unpaired T-test, (n = 41, same as in (**F**)).

DOI: https://doi.org/10.7554/eLife.27069.009

The following figure supplements are available for figure 3:

**Figure supplement 1.** Procedure and controls to deduce spike train from GCaMP signal.
DOI: https://doi.org/10.7554/eLife.27069.010

**Figure supplement 2.** Firing rate enhancement in individual neurons.
DOI: https://doi.org/10.7554/eLife.27069.011

When exposed to an AMF (22.4 kA/m at 412.5 kHz, 5 s), MNP decorated TRPV1+ neurons showed increased spiking, measured as calcium transients (***Figures 1F*** and ***3B***). The spike train of each neuron was derived from the recorded Calcium transients by convolving an estimated spike train with the recorded GCaMP6f calcium signal of a single Action Potential; then calculating the

residual between the measured and the computed calcium trace; and adjusting the estimated spike train to minimize the residual (see Materials and methods and *Figure 3—figure supplement 1*) (*Yaksi and Friedrich, 2006*). Compared to basal activity (1.8 ± 0.6 APs) during 5 s before AMF application, the spiking increased significantly to 4.5 ± 1.2 APs during the 5 s AMF (p=0.0028, unpaired T-test), and 12.1 ± 2.0 APs in the 5 s after the AMF was removed (p=0.0002, unpaired T-test) (n = 13 cells, three cultures) (*Figure 3D*, *Figure 3—figure supplement 2*). During the AMF application, the membrane temperature of MNP-decorated TRPV1$^+$ neurons increased by 2°C, as measured by the DyLight550 fluorescence (*Figure 3C*). The stimulation is very reproducible with 87.5% of the cells responding to second stimulation, following 5 min after the first.

## Latency and consistency of magnetothermal stimulation

We analyzed the consistency of activation, increase in firing rate and number of active cells, as well as the activation latency. *Figure 3E* shows the activation latency, the temporal delay between the start of field application and the first spiking event, for all recorded stimulations (79 neurons, 6 cultures, various duration of AMF). 87% of neurons spike during the first 5 s of field application, and 64% of the neurons fire already within 2 to 3 s after the field is turned on (*Figure 3E*). A histogram of the latencies of all cells exposed to 5-s field stimulation was fitted with a Poisson distribution giving an expectation value of 2.18 ± 0.17 s (*Figure 3F*). This data shows the fastest magneto-thermal stimulation to date (*Chen et al., 2015*; *Huang et al., 2010*). Next, we determined whether magneto-thermal stimulation is also capable of activating quiescent neurons, which were not already near the depolarization threshold. We defined neurons with at least one calcium transient during any 5-s period as active. At basal conditions, 53.7 ±1.6% of the neurons were active in cultures. Five-second AMF application increased the active population to 80.5 ±5.1% of neurons (*Figure 3G*). We showed that 5 s of magneto-thermal stimulation not only increases the firing rates of already active neurons (*Figure 3D*), but also increases the percentage of active cells in a population (*Figure 3G*).

## Magnetothermal genetic stimulation of the motor cortex evoked mice to run along the periphery of the arena

Next, we verified that that thermal magnetothermal genetic stimulation in the brain could be specific and sufficient to evoke precise behavior in an awake, moving animal. We aimed to evoke a response in freely moving, awake mice, that could be easily and rapidly detected. Repeatable on and off switching with reasonable latencies would demonstrate causality. Specifically, we chose several motor responses which have been previously evoked with chemogenetic or optogenetic activation.

Successful secondary motor cortex activation evoking running has been reported using optogenetics (*Gradinaru et al., 2007*). We aimed to reproduce these responses, using magnetothermal stimulation of motor cortex neurons (*Figure 4—figure supplement 1*). While some expression of TRPV1 in the rodent brain has been reported (*Basbaum et al., 2009*), robust magnetothermal genetic stimulation requires uniform, sustained over-expression of TRPV1 in the targeted neurons. We achieved TRPV1 overexpression in the mouse motor cortex by delivering adeno-associated virus serotype 5 (AAV5) as a vehicle for the TRPV1 transgene under the neuron-specific synapsin-1 promoter hSyn (AAV5-hSyn-TRPV1) by stereotactic injection (*Kügler et al., 2003*). However, AAV is too small to package a vector, encoding TRPV1 and a fluorescent protein marker. Hence, for some experiments, we also created a lentivirus carrying the genes for TRPV1 and a red fluorescent protein with nuclear targeting sequence behind an internal ribosome entry site (EF1a-TRPV1-IRES-DSRed). Both, AAV5 and lentivirus, led to robust expression in vivo, permitting stimulation of specific behavior.

Motor cortex neurons were heat sensitized, by unilaterally injecting AAV5-hSyn-TRPV1 (*Carvalho-de-Souza et al., 2015*; *Lein et al., 2007*). Two to four weeks later, 500–600 ng of antibody-conjugated MNPs were injected in the same location. (AP = 1, ML = 0.5, DV = 0.5 (all in mm)). Using immuno-histology, we confirmed virus-induced TRPV1 expression and binding of MNPs to neurons in the targeted motor cortex area (*Figure 4—figure supplement 2*).

All 6 mice in a total of 14 trials began running along the periphery of the observation arena (AMF was 7.5 kA/m and 570 kHz; *Figure 4A,B*, *Figure 4—figure supplement 3*). To illustrate the contrast between the induced and the resting behaviors, the track of the head (color marked for automated object recognition) of a representative mouse is shown in *Figure 4C*. The running along the

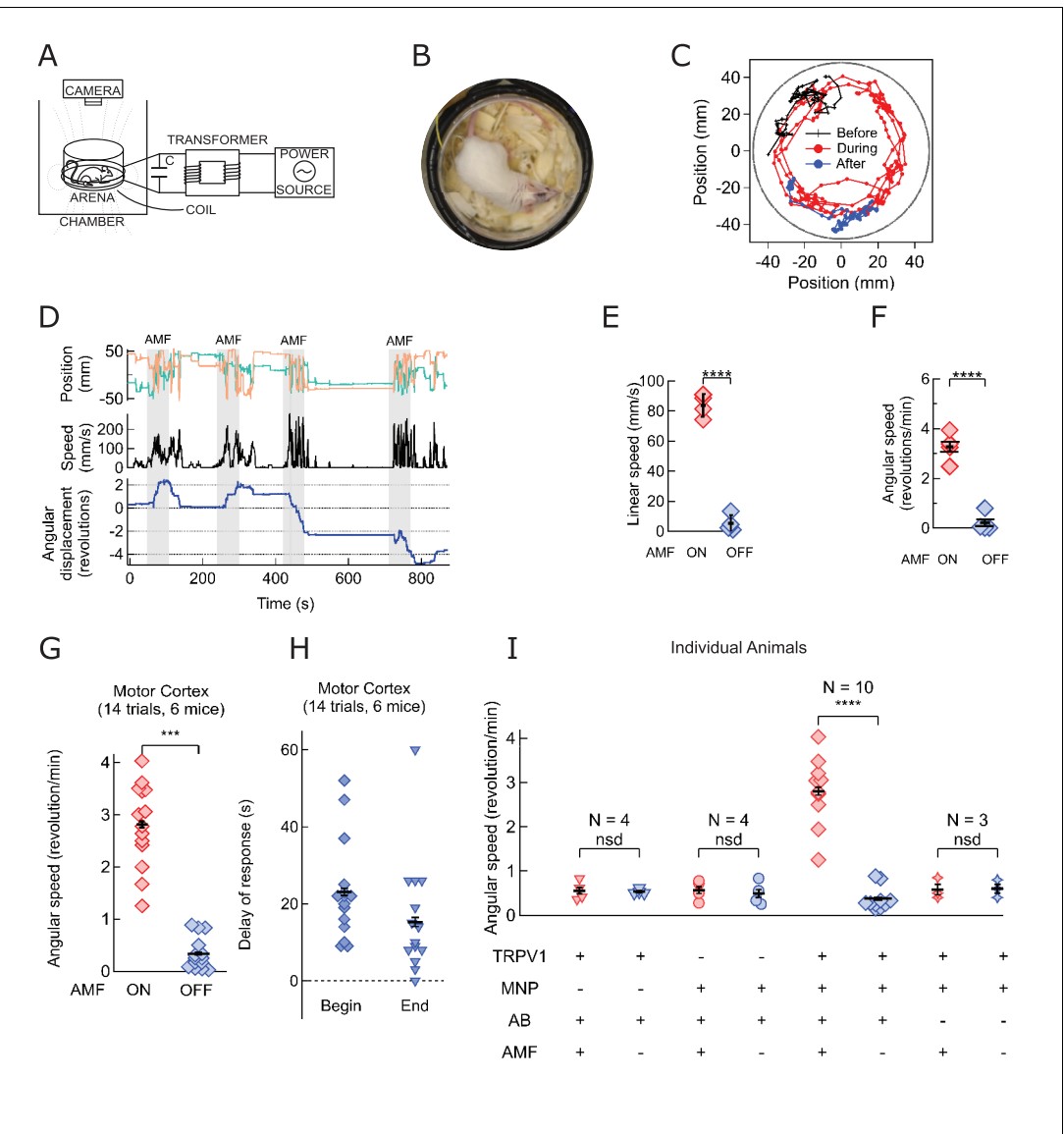

**Figure 4.** Magnetothermal neurostimulation in the motor cortex elicits fast scurries around the arena. (**A**) Experimental set-up for in vivo magnetothermal stimulation of motor behavior in awake mice. A water-cooled two-turn coil around the arena generated the AMF. The AMF in the coil was powered by the same system as for the in vitro experiments. An overhead camera was used to record the mouse's behavior in the arena. (**B**) Photograph of mouse in the observation arena; two-turn water-cooled coil visible as black ring. (**C**) Representative trajectory recorded from a mouse stimulated in the motor cortex before (black), during (red), and after (blue) field application (each 1 min long). The black circular border denotes the actual boundary of the chamber. (Also see *Figure 4—figure supplement 1* for injection locations; and *Videos 1* and *2*). (**D**) (Top) Position of the mouse's head, (x green; y orange) measured taking the center of the arena to be the origin. (Middle) Black trace shows the linear speeds of the mouse. Speed markedly increases during all AMF applications (Grey bars). After each AMF application, the mouse slows down regular exploratory motion (Bottom). Total turns made by the mouse was tracked versus time. Counter-clockwise angular changes were counted as a positive change in angles. During AMF application, the mouse turned unilaterally significantly more than between the AMF applications. (**E**) Comparison of linear speed of this TRPV1$^+$ / MNP$^+$ mouse, injected in the motor cortex, with and without AMF. The average linear speed increased 16-fold after AMF application, from 5.3 ± 2.75 mm/s, before AMF, to 83.8 ± 3.75 mm/s, during AMF (one mouse, four trials, the error bars are smaller than the symbols; p=5.9·10$^{-6}$, 95% confidence intervals [0.9, 9.7] and [77.8, 89.8] mm/s). (**F**) Comparison of the angular speed from the same mouse as in (**E**). The angular speed during the AMF was 3.27 ± 0.30 rev/min versus 0.21 ± 0.19 rev/min between the AMF applications (one mouse, four trials, error bars are smaller than symbols; p=0.0003, 95% confidence intervals [−0.10, 0.53] and [2.78, 3.74] rev/min). (**G**) Comparison of angular velocity, or speed of circling the arena, measured for all TRPV1$^+$ /

*Figure 4 continued on next page*

*Figure 4 continued*

MNP$^+$ mice, injected in the motor cortex, with or without AMF. The speed of circling the arena in revolutions per minute increased 8-fold with AMF, from to 0.34 ± 0.08 rev/min to 2.81 ± 0.20 rev/min (6 mice, 14 trials; p=5.2·10$^{-9}$; 95% confidence intervals [0.29, 0.39] and [2.69, 2.93] rev/min). (H) Latency of behavioral response onset and end after turning field on and off, respectively (n = 21). (I) Speed of rotation for control and experiment animals (independent mice): *Control1*: PBS instead of MNP injected (TRPV1$^+$ / MNP$^-$, n = 4; *Video 3* ); *Control2*: PBS instead of virus injected (TRPV1$^-$ / MNP$^+$, n = 4; *Video 4*); and *Experiment* (TRPV1$^+$ / MNP$^+$). There was no significant difference in the observed speeds with or without AMF (AMF$^+$ and AMF$^-$, respectively) in the control cases. With TRPV1$^+$ / MNP$^+$, the mice exhibited a highly significant increase in speed with AMF, 3.17 ± 0.17 rev/min, as compared to the mice without AMF application, 0.42 ± 0.07 rev/min (n = 10, p=1.1·10$^{-5}$; unpaired T-test; 95% confidence intervals [0.32, 0.52] and [2.94, 3.40] rev/min). In the experiments and controls 1 and 2 and MNP were injected with A2B5 antibody (AB). *Control3*: MNP without antibody injected (TRPV1$^+$ / MNP$^+$, AB$^-$, n = 3).
DOI: https://doi.org/10.7554/eLife.27069.014

The following figure supplements are available for figure 4:

**Figure supplement 1.** (Top) Angular displacement (negative CW, positive CCW) of mice stimulated magnetothermally in the motor cortex (7 trials, 3 mice).
DOI: https://doi.org/10.7554/eLife.27069.015

**Figure supplement 2.** Virus-induced TRPV1 over-expression and membrane-bound MNP labeling of neurons in the Motor Cortex.
DOI: https://doi.org/10.7554/eLife.27069.016

**Figure supplement 3.** An example for a mouse injected with virus and MNPs but without the anti-A2B5 antibody.
DOI: https://doi.org/10.7554/eLife.27069.017

**Figure supplement 4.** Repeatability of stimulation in motor cortex.
DOI: https://doi.org/10.7554/eLife.27069.018

**Figure supplement 5.** Magnetic field distribution across experimental arena.
DOI: https://doi.org/10.7554/eLife.27069.019

periphery of the circular arena, recorded during field application (shown in red) (*Video 1*) is in sharp contrast to the exploratory/resting tracks, observed in the absence of AMF. Control data acquired from the same mouse before MNP injection, and again between each trial showed randomly oriented slower locomotion, not confined to the periphery (*Video 2*). Shortly after the AMF application began, the mouse initiated running, which slowed down quickly after the AMF was removed (*Figure 4D*). The linear running speed of the mouse increased from 5.3 ± 2.75 mm/s to 83.8 ± 3.75 mm/s (*Figure 4E*; one mouse, four trials, p=5.9·10$^{-6}$, 95% C.I. [0.9, 9.7] and [77.8, 89.8] mm/s). The angular speed increased from

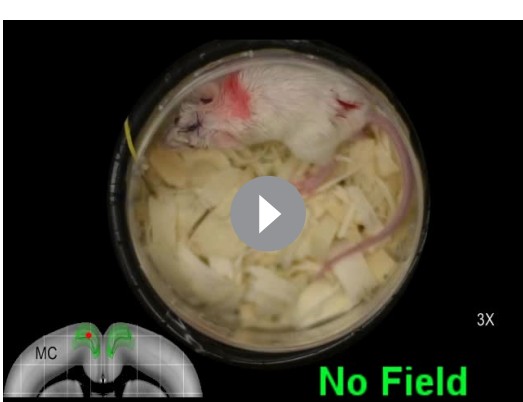

**Video 1.** Ambulatory response generated by secondary motor cortex stimulation. Unilateral stimulation in the motor cortex evokes running with some delay. The ambulatory response persists during the 60 s long field application and the stimulation can be repeated within minutes. The video shows one continuous experiment with four field, which each lead to a long run, typically 3–4 times around the arena. Video speed is 3x accelerated, MNP injection and application of the field marked within the video frames.
DOI: https://doi.org/10.7554/eLife.27069.012

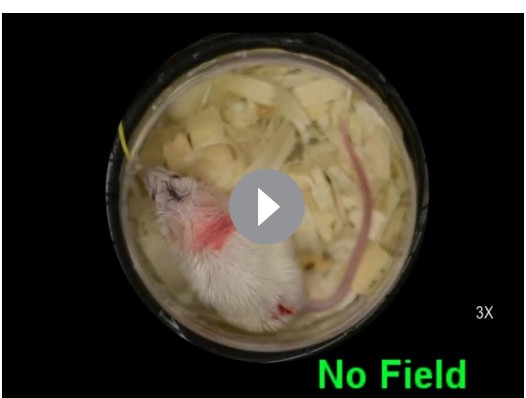

**Video 2.** Control video of the same mouse without any field application.
DOI: https://doi.org/10.7554/eLife.27069.013

0.21 ± 0.19 rev/min to 3.27 ± 0.30 rev/min (*Figure 4F*; p=0.0003). Over all 14 trials in 6 mice, the angular speed was significantly faster during AMF applications, 2.81 ± 0.20 rev/min, as compared to 0.34 ± 0.08 rev/min during the rest periods between the fields (*Figure 4G*; p=5.2·10$^{-9}$; 95% C.I. [0.29, 0.39] and [2.69, 2.93] rev/min). *Figure 4—figure supplement 1* shows seven individual responses to the simulation (four mice). In all trials, we observed a strong running response, but some mice changed directions occasionally. Hence, we plot the angular distance travelled, in addition to the angular displacement.

Repeat stimulation of the same animal evoked increased locomotion reliably in each trial. *Figure 4D* shows four 1-min field applications within a 15-min trial. Repeated sessions on the same mice demonstrate that the sensitization persists over several days (*Figure 4—figure supplement 4*). There is a tight temporal correlation between AMF heating and the observed behavior: Each time, shortly after the AMF is applied, the mouse begins to run. The stimulated ambulation persists throughout the duration of the field application and ends shortly after the AMF is removed. For an AMF strength of 7.5 kA/m rms at 570 kHz, the behavioral response began 22.8 ± 2.6 s (n = 14, 6 mice, *Figure 4H*) after the beginning of the AMF. This latency of behavior onset is consistent with the fast in vitro response, because heating depends on the square of the AMF strength, which for the in vivo case was about one third of that in in vitro experiments. Performing AMF heating on MNP decorated HEK cells showed that lowering the field strength by a third increased the required time to reach a two-degree increase to 20 s (*Figure 2—figure supplement 1*). The delay is comparable to optogenetic stimulations in the motor cortex, also using virus delivered activator genes (*Airan et al., 2009*). After the AMF is terminated, the ambulation behavior continued for 14.7 ± 2.5 s, likely the time required to cool the membrane-bound MNPs (*Figure 4H*). Using membrane-bound nanoparticles for magnetothermal neuro-stimulation ensures fast on and off kinetics because only a miniscule volume of space is heated and cooled.

Mice injected with only AAV5-hSyn-TRPV1 virus or only MNPs showed no response to AMF application (*Videos 3* and *4* respectively) demonstrating that neither MNP injection nor TRPV1 expression alone is sufficient to evoke the response. Observable stimulation occurred only when AMF was applied in the presence of the temperature-gated ion channel, TRPV1, and the energy transducing MNPs in the same brain locations. For animals lacking either TRPV1 overexpression or MNP, AMF application caused no significant deviation from baseline rotation speedEquation (four individual mice for each, *Figure 4I*). On the other hand, in all mice injected with both MNP and TRPV1 at the same brain location, angular speed increased significantly with AMF, 3.17 ± 0.17 rev/min compared to the baseline speed of 0.42 ± 0.07 rev/min (n = 10, p=1.1·10$^{-5}$; unpaired T-test; 95% C.I. [0.32, 0.52] and [2.94, 3.40] rev/min) (*Figure 4I*). Membrane targeting the MNPs is essential as animals injected with virus and MNPs without the anti-A2B5 antibody were insensitive to the AMF application (*Figure 4I*, *Figure 4—figure supplement 3*).

## Deep brain magnetothermal genetic stimulations in striatum causes rotation around body axis

Next, we investigated the effectiveness of magnetothermal genetic stimulation in the striatum, a deep brain region. Chemogenetic activation of the caudate putamen nuclei in the striatum has previously been shown to evoke increased locomotion in the form of rotation around the body axis (*Arenkiel et al., 2008*; *Kreitzer and Malenka, 2008*). To replicate these results with magnetothermal stimulation, we injected AAV5-hSyn-TRPV1 virus and MNPs into the caudate putamen nuclei in the dorsal striatum, 3 mm deep (AP = 0, ML = 2.3 from bregma)(*Li et al., 2015*; *Oh et al., 2014*). Immuno-histology confirmed successful TRPV1 over-expression, MNP injection to the target (*Figure 5—figure supplements 1,2*), and good overlap of both (*Figure 5—figure supplement 3*. During AMF application, the mouse

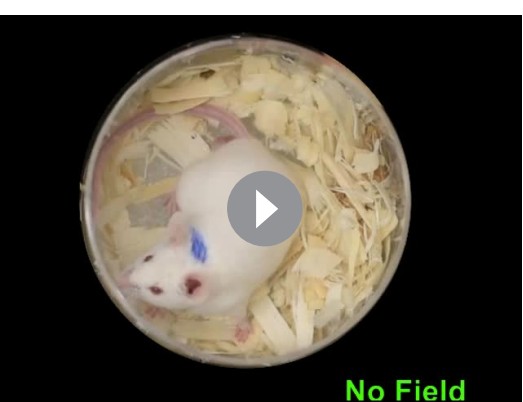

**Video 3.** Control video of a mouse injected with virus but not MNP.
DOI: https://doi.org/10.7554/eLife.27069.020

turns around its body axis, so that its head tracks circles of 22 ± 5 mm radius (*Figure 5A-C*, *Video 3*). These turns were very regular and quick (19.3 ± 2.2 mm/s; *Figure 5D*), resulting in 4.7 ± 0.8 revolutions per minute (*Figure 5E*). This behavior was very different from the running motion observed during motor cortex activation, where the mice ran at speeds of 83.8 ± 3.75 mm/s, close to the edge of the arena, which has a radius of 50 mm. In this case, the turn radii of the circles made by the mice were considerably smaller, and the linear speed lower. The turning motion evoked by stimulating in the striatum was very reproducible from trial to trial and across animals (*Figures 5F*, 12 trials, 4 animals). The mice turned contralateral to the injection in 11 out of 12 trials and did never reverse the direction of turning after initiating the motion.

The average response of four striatum targeted mice in 12 trials was a 6-fold increase in angular speed during AMF activation, 2.86 ± 0.19 rev/min compared to a baseline rate of 0.47 ± 0.09 rev/min (*Figure 5G*; p=3.4·10$^{-7}$; unpaired T-test; 95% C.I. [0.42, 0.56] and [2.71, 3.01] rev/min). The observed response is consistent with a prior report of dorsal striatum stimulation by capsaicin activation of TRPV1 channels at the same brain coordinates (*Arenkiel et al., 2008*).

In addition to multiple stimulation trial during one session, we performed several sessions successfully over 54 hr (*Figure 5—figure supplement 4*).

## Magnetothermal genetic deep brain stimulation at the ridge between dorsal and ventral striatum causes freezing of gait

To evoke magnetothermally a response that does not involve increased locomotion, we aimed to demonstrate stimulation freezing of gait. The Deisseroth team showed that optogenetic stimulation near the ridge between dorsal and ventral striatum inhibits voluntary locomotion (*Gradinaru et al., 2009*; *Kravitz et al., 2010*). Therefore, we took advantage of the fact that the stimulation depth is only limited by brain region accessibility in terms of virus and MNP delivery, and injected AAV5-hSyn-TRPV1 virus and MNPs at the ridge between dorsal and ventral striatum (AP = 0.01, ML = 2.3) (*Figure 6—figure supplement 1*). Application of AMF caused freezing of gait (FOG), characterized by the inhibition of locomotion, with all four paws locked in place while the mouse was still able to move its head in all directions, neck onward. FOG is common in patients with advanced Parkinson disease (*Moore et al., 2008*). *Videos 6* and *7*, *Figure 6A–C* shows tracks of the mouse neck. Ambulatory activities diminished markedly during stimulation, which can also be seen from the paw tracks (*Figure 6—figure supplement 2*). Apart from FOG, there was an excessive outward stretching of the limbs along with the inability of the front extremities to follow the head motion (*Figure 6D*). During AMF application, the average linear speed decreased significantly, from 17.7 ± 2.2 mm/s before stimulation to essentially rest, 1.04 ± 0.13 mm/s (*Figure 6C*; two mice, six trials; p=0.0006; unpaired T-test; 95% C.I. [15.4, 20.1] and [0.90, 1.18] mm/s). The tracked speed during the AMF was indistinguishable from that of a fixed marker on the arena floor (0.52 ± 0.11 mm/s).

## Discussion

Our work demonstrates that magnetothermal genetic stimulation can effectively activate specific neuronal circuits in awake, freely moving mice to evoke defined behaviors quickly, robustly and repeatedly. Three different circuits and behaviors have been stimulated magnetothermally and compared to prior opto- or chemogenetic stimulation of the same circuits: running, in response to motor cortex stimulation (prior optogenetic work: [*Gradinaru et al., 2009*]); rotation around the body axis, after stimulation deep in the striatum (prior chemogenetic work: (*Arenkiel et al., 2008*); and freezing of gait, in response to stimulation on ridge between dorsal and ventral striatum (prior optogenetic work: [*Kravitz et al., 2010*]). We successfully evoked opposing behaviors, running or freezing of gait by targeting the dorsal striatum or the ridge between the dorsal and ventral striatum, respectively, locations in the brain only 1 mm apart. This clearly demonstrates that the evoked behavior is the result of stimulating a specific neuronal circuit, and that the heat generated by the MNPs in response to the AMF is sufficiently localized to the neurons targeted.

Magnetothermal genetic stimulation leads rapidly to the targeted motor behavior, observable within 15–20 s of AMF activation. This brief delay is not an intrinsic limit of the technique, but rather the result of a weak average field at the site of the mouse head (*Figure 4—figure supplement 5*). As the heating rate is proportional to the square of magnetic field strength, the threefold weaker field caused approximately nine times longer latency in the mouse than observed in the in vitro

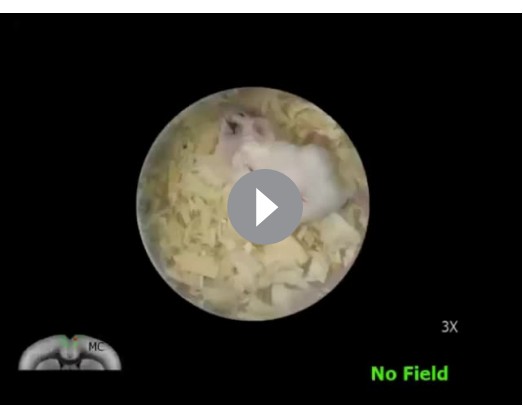

**Video 4.** Control video of a mouse injected with MNP but no virus.
DOI: https://doi.org/10.7554/eLife.27069.021

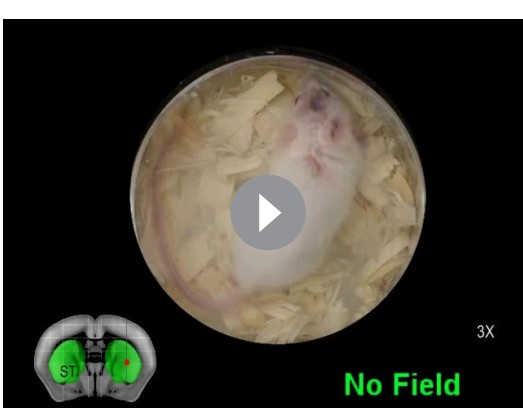

**Video 5.** Body-axis turning caused by local stimulation in the dorsal striatum. The video shows the mouse's response to field, resulting from stimulation of striatum. The response is repeatable, as shown here in one continuous recoding with two trials. The mouse makes two complete and uninterrupted ipsilateral turns during the first field application and one complete ipsi - and contra - lateral turn each during the second field. The turn radii are not limited by the arena boundary. Video playback speed 3X; injection site and field times are shown as overlays.
DOI: https://doi.org/10.7554/eLife.27069.022

studies. It is technically feasible to generate AMFs large enough to reduce the latency for magnetothermal stimulation of behavior in mice to times only marginally slower than electrophysiological or optogenetic stimulation, and significantly faster than chemical stimulation using DREADDs (*Armbruster et al., 2007*). Likewise, our results show that the evoked locomotor behavior diminished quickly within 15 s after removal of the field. Tight temporal on- and off-control is vital for precise correlation of circuit stimulation with behavioral observation, and thus far had only been achieved using optogenetics, but not magneto- or chemogenetics.

The short on- and off-time delays of magnetothermal genetic stimulation permit multiple simulations in short observation times. In our experiments, three to four one-minute stimulations were possible within a 10–15 min experiment. Such repeatability is crucial for obtaining statistically significant behavioral data. Other temporal profiles of stimulation are easily achieved, permitting studies of memory formation, or with variations in the assay. Most importantly, stimulation from several seconds to many minutes are easily achievable, filling an important gap between faster optogenetics and slower chemogenetics.

In this study, MNPs were bound via the A2B5 antibody, which indiscriminately recognizes all glycosylated membrane proteins. It yielded the highest nanoparticle density on the cell surface and accordingly, achieved the most rapid heating. However, we also successfully used anti-CD56 (Thy-1), anti-CD90 (NCAM) for endogenous membrane proteins, as well as anti-GFP to target overexpressing GFP-labeled membrane proteins. Magnetothermal genetics in tandem with membrane-bound nanoparticles provides three major benefits: First, it provides improved localization and neuron-specific targeting. The stimulation region is spatially better confined as the antibody binding limits the distance that the MNP diffuse into the brain (*Figure 5—figure supplement 5*). Also, only cell types with the specific membrane receptor targeted by the antibody bind the nanoparticles (*Figure 5—figure supplement 6*). Second, the heating is restricted to the vicinity of the membrane. This enables orthogonal targeting in which the TRPV1 channel is expressed in one specific cell type and the MNPs are targeted to a second cell type, and magnetothermal stimulation would only occur at their synapse. Third, membrane targeting provides improved temporal resolution because only a minuscule volume near the membrane is heated which cools off immediately after the AMF is removed. This also delivers less overall heat to the brain. Lastly, we obtained robust activation with as little as 500 ng of nanoparticles, which is 200-fold lower than the volume of nanoparticle suspension for of brain neuron stimulation in anesthetized mice (*Chen et al., 2015*).

One concern for magnetothermal stimulation may be that the MNPs either directly or via the membrane heating induce cell death. We successfully repeated stimulation of the same animals for seven to eight trials over a few days

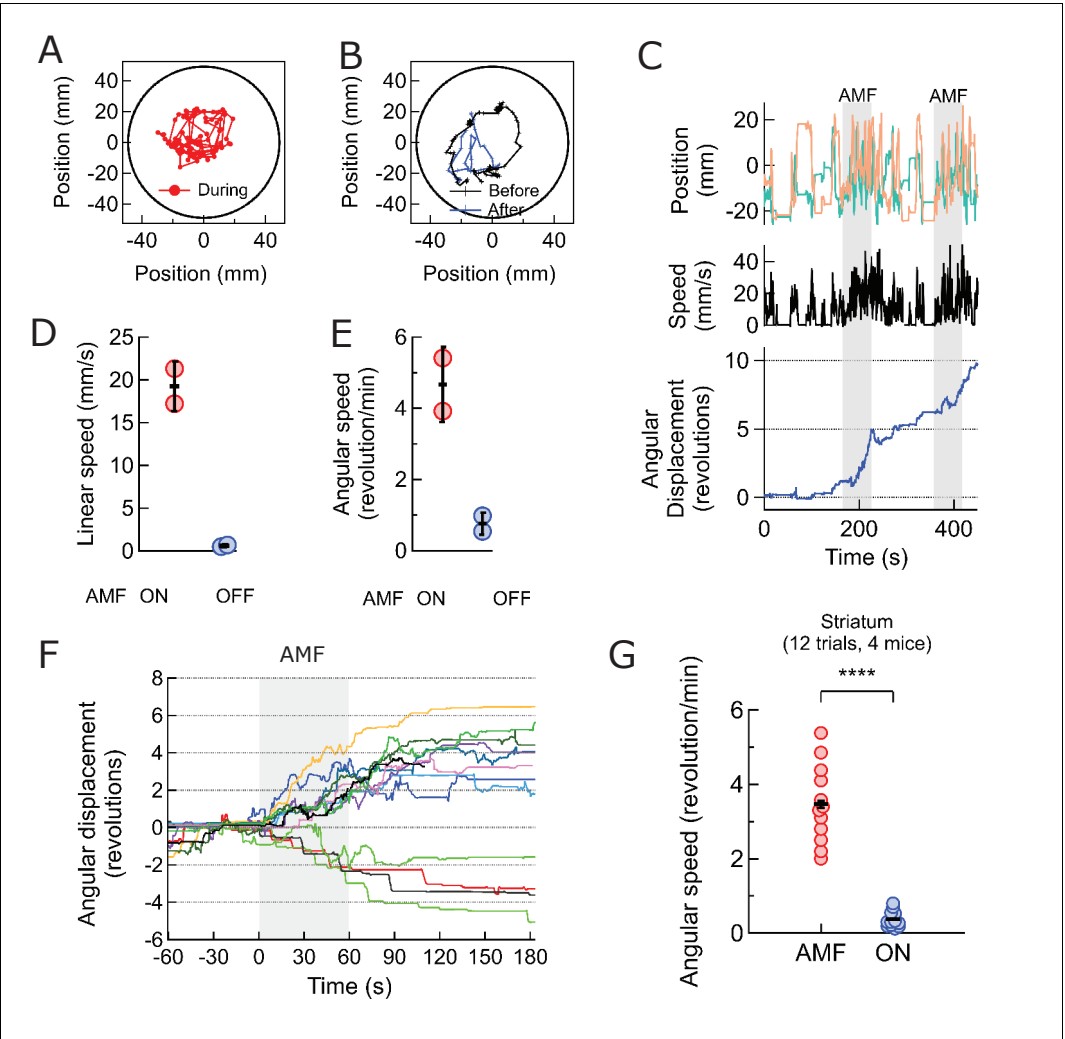

**Figure 5.** Magnetothermal neurostimulation in the striatum elicits rapid rotations around body axis. (**A**) Trajectory of a representative TRPV1$^+$/MNP$^+$ mouse, injected in the striatum, during AMF application (1 min long). The mouse remained near the center of the arena, as it turned unilaterally around its body axis (in contrast with the motor cortex injected mouse). Black circle represents the actual arena boundary. (**B**) Two one-minute trajectories of the same mouse without AMF (black: before AMF application, blue: after AMF application). The mouse carried out regular resting/exploratory activities, turning five times less than that with AMF (Also see **Video 6**). (**C**) (Top) Trajectory (x,y positions, x green; y orange) of the mouse shown in part (**A**) and in **Video 5**. One minute long AMF was applied twice (gray bar). (Middle) Linear speed in mm/s of the same mouse calculated from the trajectory. (Bottom) Plot of the angular displacement, in revolutions, calculated from the trajectory. The mouse turns unilaterally counterclockwise (to the left) during the AMF applications. (**D**) , (**E**) Comparison between linear and angular speeds of the same mouse with (red) and without (blue) AMF. (**F**) Records of 12 stimulation trials in four mice. The motion is tracked and the angular displacement recorded (negative angular displacement is CW, while positive is CCW). In AMF application, the mouse only turns in one direction, typically for 3 to 4 turns, and in 11 out of 12 cases it turns contra-lateral to the injected brain half. (**G**) Comparison of angular velocity, or turning speed, in revolutions per minute measured for all TRPV1+/MNP + mice, injected in the striatum: angular speed during AMF, 3.46 ± 1.04 rev/min, was 5.8-fold the angular speed without AMF, 0.37 ± 0.22 rev/min (4 mice, 12 trials, p=3.74·10$^{-7}$; unpaired T-test; 95% confidence intervals [0.33, 0.41] and [3.27, 3.65] rev/min, mean ± s.d.).
DOI: https://doi.org/10.7554/eLife.27069.023

The following video and figure supplements are available for figure 5:

**Figure supplement 1.** Confirmation of targeted region in the Striatum.
DOI: https://doi.org/10.7554/eLife.27069.024

**Figure supplement 2.** Co-localization of virus induced channel expression and MNP binding in the striatum.
DOI: https://doi.org/10.7554/eLife.27069.025

*Figure 5 continued on next page*

*Figure 5 continued*

**Figure supplement 3.** Virus-induced TRPV1 over-expression in the Striatum.
DOI: https://doi.org/10.7554/eLife.27069.026
**Figure supplement 4.** Repeatability of stimulation in striatum.
DOI: https://doi.org/10.7554/eLife.27069.027
**Figure supplement 5.** Antibody-assisted targeting confines MNP spread.
DOI: https://doi.org/10.7554/eLife.27069.028
**Figure supplement 6.** Antibody enhances membrane targeting of MNPs.
DOI: https://doi.org/10.7554/eLife.27069.029
**Figure supplement 7.** Neuronal cells health after multiple AMF exposures.
DOI: https://doi.org/10.7554/eLife.27069.030
**Figure 5—video 1.** Cellular resolution confocal z-sections of MNP with anti-A2B5 antibodies binding neurons in the brain. (example 1)
DOI: https://doi.org/10.7554/eLife.27069.031
**Figure 5—video 2.** Cellular resolution confocal z-sections of MNP with anti-A2B5 antibodies binding neurons in the brain. (example 2)
DOI: https://doi.org/10.7554/eLife.27069.032
**Figure 5—video 3.** Cellular resolution confocal z-sections of MNP with anti-A2B5 antibodies binding neurons in the brain (example 3).
DOI: https://doi.org/10.7554/eLife.27069.033
**Figure 5—video 4.** Cellular resolution confocal z-sections of MNP with anti-A2B5 antibodies binding neurons in the brain (example 4).
DOI: https://doi.org/10.7554/eLife.27069.034

without observing any change in normal or evoked behavior. Also, histology of brains after 20 one-min AMF applications, spread over three sessions during a 24 hr period, showed intact MNP decorated neurons (*Figure 5—figure supplement 7*). Hence, if there is any damage, it is undetectable in histology and behavior. Thermal damage can be avoided by choosing the field strength to be sufficiently small so that the cooling to the environment limits the possible temperature rise to a few degrees.

As magnetothermal genetic stimulation does not require a physical connection to the animal, it offers unprecedented freedom in designing novel behavioral assays. For instance, several animals could share the stimulation arena simultaneously, and only animals injected with MNPs would be affected by the AMF. This provides the ability to study stimulated and unaffected animals simultaneously, either as controls or to study their interaction. Absence of a tether or implant also has the advantage that animals do not carry any external marker, which may interfere with normal social group behavior. Additionally, exclusive to magnetothermal genetic stimulation, stimulation only occurs when the animal is within the magnetic field. Hence, adapting the field geometry allows controlling the stimulation timing and condition. This is ideal for space or location dependent assays such as multi chamber place preference assays when the animals are to be stimulated only when in one chamber.

One limitation of the current implementation of magnetothermal genetic stimulation is the power required to generate high AMFs over a large volume. In this study, a 7.5 kW power source produced the AMF over a volume of

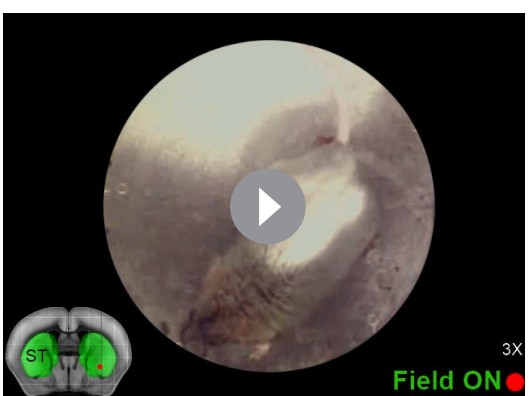

**Video 6.** Freezing of gait caused by local stimulation in the deep striatum. The mouse was injected 4.1 mm deep in the striatum, near the ridge between Dorsal and Ventral striatum. Upon field application, the limbs and tail of the mouse are rendered motionless, while the rest of the body, including the neck retains normal mobility. Sometime past the stimulation, the mouse regains control over the limbs and moves freely again. Video playback speed in 3X the recording speed. Injection site in the brain and field times are shown as overlays.
DOI: https://doi.org/10.7554/eLife.27069.035

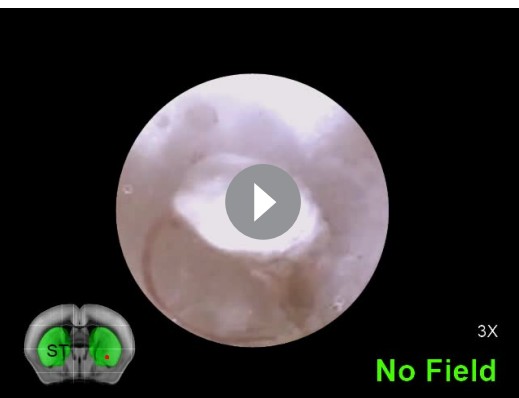

**Video 7.** Control video of Freezing of gait.
DOI: https://doi.org/10.7554/eLife.27069.036

300 cm³. However, stronger power sources are available to produce AMF in larger volumes, sufficient to cover the entire brain of non-human primates as well as humans. Magneto-thermal stimulation may provide an alternative to optogenetics for deep brain stimulation in non-human primates. First, large brains move significantly relative to the skull, potentially displacing electrodes and glass fibers from their original target. As the transducer for magneto-thermal stimulation is attached to the cells and not the skull, the stimulation location remains with the originally targeted cells. Second, activation of behavior in non-human primates may require coordinated activation of several distinct brain regions and each region may be larger than the volume covered by an optical fiber (*Han, 2012*; *Inoue et al., 2015*). Magneto-thermal genetic stimulation can stimulate multiple distinct brain locations easily, as the delivery of the MNP is minimally invasive and the magnetic field covers the entire volume. The actual stimulated volume can be controlled by the MNP amount and rate of injection. Therefore, magnetothermal genetic stimulation is well suited for deep brain stimulation in non-human primates.

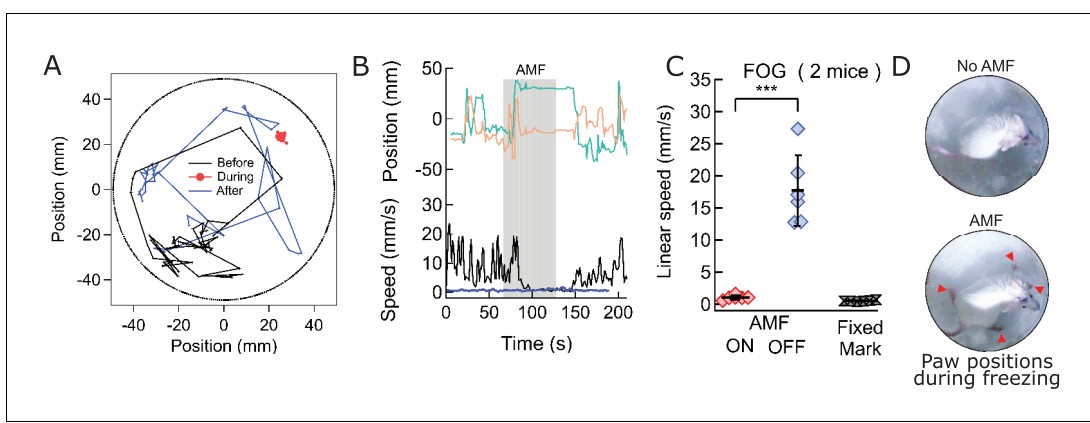

**Figure 6.** Magnetothermal stimulation near the ridge between ventral and dorsal striatum caused freezing of gait (FOG). (**A**) Track of mouse stimulated in deep striatum before (black), during (red), and after (blue) AMF application (each 1 min (Also see *Figure 6—figure supplement 2*, *Videos 4* and *5*). (**B**) X,Y-position and linear speed of same animal tracked at the neck. The speed is compared to a fixed reference point in the sample chamber (blue line; Grey bar indicates the field duration). (**C**) Scatter plot, showing the response of two mice with AMF. The average speed without AMF (17.7 ± 2.2 mm/s) was significantly faster than that with AMF (1.04 ± 0.13 mm/s) (two mice, six trials; p=0.0006; unpaired T-test; 95% confidence intervals [15.4, 20.1] and [0.90, 1.18] mm/s). The speed during AMF was very similar to the speed of fixed reference marker (0.52 ± 0.11 mm/s). (**D**) Still picture of same before (top) and during (bottom) field. The red arrowheads indicate paw position during stimulated freezing of gait (FOG). During the AMF, the mouse is able to move its head but not control the limbs which end up stretching farther apart than normal and are unable to follow the head's motion. In the absence of AMF, ambulation is normal and front paws are positioned to both sides of the head positon and hind paws underneath the animal's body.
DOI: https://doi.org/10.7554/eLife.27069.037

The following figure supplements are available for figure 6:

**Figure supplement 1.** Targeted Region in the Striatum.
DOI: https://doi.org/10.7554/eLife.27069.038

**Figure supplement 2.** Paw Location Data for Freezing of Gait Response.
DOI: https://doi.org/10.7554/eLife.27069.039

Just as for other genetic techniques, one hurdle to application in humans will be safety concerns around injecting virus and nanoparticles into the brain. However, the body's reaction to nanoparticle injections has been extensively studied and can be minimized by using a biologically compatible nanoparticle surface coating. The invasive brain surgery currently used to inject the MNP may soon be replaced by MNP delivery via the bloodstream as recent research has shown, that focused ultrasound can transiently make the blood-brain barrier permeable to virus particles (*Price, 2015*). Overall, magnetothermal neurostimulation offers a series of advantages for deep brain stimulation or silencing in non-human primates and primates.

## Materials and methods

### Magnetic field generation

Alternating magnetic fields were generated by water-cooled coils, driven by a 7.5 kW power generator from MSI Automation. The instrument is separated into two units (*Figure 1C*, *Figure 1—figure supplement 4*). The power source (also the control unit) was some distance away from the microscope while the head stage, on which the hyperthermia coil was mounted, was placed over the microscope stage through an opening in the incubator. The coil was placed directly over the sample chamber. Circulating water cooled any resistive element in the systems. Magnetic field for all in-vitro experiments were generated using a 5 mm $\Phi$, five turn copper coil. Field measurements at the samples' site were done separately using a Fluxtrol Magnetic Probe. For in vivo studies, the same instrument was used to power a 10 mm $\Phi$, two-turn water cooled copper coil (*Figure 4A*, *Figure 4—figure supplement 5*).

### Magnetic nanoparticle preparation

Superparamagnetic core-shell Co-Mn-Ferrite nanoparticles (MNP) with a diameter of 12.5 ± 1.0 nm (*Figure 1D*) were synthesized and characterized according to published methods(*Zhang et al., 2015a*). The MNP were transferred to water by coating with PMA (poly-isobutylene-maleic anhydride) following published protocols(Lin et al., 2008). The PMA-coated MNP were functionalized by covalently attaching fluorescently labeled Neutravidin, which could then bind to a biotinylated antibody.

### Experimental and theoretical magnetic nanoparticle characterization

Magnetization data for Horse spleen ferritin was generously shared with us by Dr. I. Orue (SGIker, Universidad del País Vasco UPV/EHU, 48940 Leioa, Spain). Magnetization data for the synthesized core-shell MNP, Co-Mn-Ferrite nanoparticles (MNP) was also measured by Dr. I. Orue; *Figure 1—figure supplement 1A* . For calculation of magnetic dipole interaction energy (*Figure 1—figure supplement 1*), spherical MNPs were assumed to be point dipoles, with moments aligned perpendicular to the dipole axis. The interaction energy, then is given by

$$U(r) = \frac{\mu_0}{4\pi r^3} \cdot \left(2\mu^2\right)$$

where μ is the magnetic moment of individual nanoparticles separated by a distance r.

For calculation of SLP values (*Figure 1—figure supplement 1*), a saturation magnetization value of 340 μB was used for horse spleen ferritin. Anisotropy constant was obtained from reported values (*García-Prieto et al., 2016*). Saturation magnetization and anisotropy constant for core-shell MNPs were obtained from previous measurements (*Zhang et al., 2015b*). The permissible magnetic field strength for each frequency points were calculated using the accepted limit of field attenuation (and ensuing eddy current heating in tissues) (*Atkinson et al., 1984*; *Chen et al., 2015*). Power delivered in W/g of particles is then given by:

$$P_{particle} = \pi \mu_0 \chi_0 H_0^2 f \frac{2\pi f \tau}{1 + (2\pi f \tau)^2}$$

Here, the magnetic susceptibility $\chi_0$ follows Langevin's equation and is a function of both the magnetic field amplitude H and T (*Rosensweig, 2002*).

## Molecular thermometry

To measure the local heating near the nanoparticle decorated cell surface, we used the fact that the fluorescence lifetime as well as intensity of molecular fluorophores and fluorescent proteins decrease with increasing temperature (*Figure 1—figure supplement 3*).

## Fluorescent labeling of neutravidin and binding to MNPs

Neutravidin (31000, Thermo Scientific), dissolved in Phosphate-buffered saline (PBS) at 2 mg/ml (pH 7.34), was labeled with DyLightTM 550 NHS ester dye (62262, Life Technologies), or Alexa 647 dye (A37573, Life technologies) by mixing Neutravidin stock solution with five times molar excess of the fluorophore, and incubated in dark for 2 hr. Excess dye (molecular weight, under 3 kDa) was removed using a Microcon 10 kDa centrifugal filter (EMD Millipore), 8000 r.c.f. Carboxylic groups on surface of the PMA-MNP (1 mg/ml) were activated with 0.5 mg/ml of EDC [1-ethyl-3-(3-dimethylami-nopropyl) carbodiimide hydrochloride] (22980, Life technologies) and 1.4 mg/ml of Sulfo-NHS (N-hydroxysulfosuccinimide) (24510, Life technologies). Activated MNPs were re-suspended in PBS, and added dropwise to Neutravidin -Dylight 550 or -Alexa 647, in 10 times molar excess. After reacting for two hours in the dark, the reaction was quenched with 10 mM Hydroxylamine buffer. Unbound Neutravidin was removed by washing MNP in 300 kDa cut-off centrifuge filters (8000 r.c.f).

## Fluorescence microscopy

### Image acquisition and imaging conditions

Fluorescence intensities of GCaMP6f as a measure for $Ca^{2+}$ levels, and DyLight 550 dye (Life Technologies, Carlsbad, CA) as molecular thermometer, were recorded using an inverted epifluorescence microscope (ZEISS AXIO OBSERVER A1.0m). Cells grown on 12 mm glass coverslip were secured in a non-conducting chamber (ALA MS-512DWPW) and placed on the XY stage of the microscope. A custom-built environmental incubation chamber enclosed the microscope. Temperature inside the chamber was set before the start of the experiment and kept stable at 33.0 ± 1.0°C. The fluorophores were excited using a HBO 200 mercury short arc lamp with appropriate filters. The emission was collected with a 40x NA 0.75 Zeiss objective lens and recorded with an Andor NEO sCMOS camera controlled by micromanager software (*Edelstein et al., 2014*). Continuous videos were acquired with an exposure time of 100 ms at 2 × 2 binning, and saved as stack of uncompressed. TIFF files. Uniblitz electronics VS14S2T0 shutter, driven by Uniblitz VCM-D1 driver was used to synchronize excitation light illumination with camera acquisition times via micromanager software.

### Microscopy in alternating magnetic field

The alternating magnetic fields induce eddy currents in the metallic microscope objective, heating it up. The thermal expansion of the objective lens may cause optical distortions and focus changes. To correct these in real time, we used a fast, customized piezoelectric autofocus system (Motion X Corporation, 780 nm laser interferometer based). The autofocus laser beam was coupled in the optical path of the microscope by a dichroic mirror, focused through the objective lens onto the coverslip bottom and reflected back into the interferometer (*Figure 1—figure supplement 4*). Therefore, it not only detects and corrects changes in separation between coverslip and objective lens but also internal optical changes.

## Labeling cells with MNP

### HEK 293 T cells

HEK cells growing on 12-mm glass coverslip, 50% confluent were co-transfected with plasmid expressing Biotin acceptor arm AP-CFP-TM and plasmids BirA enzyme (p-DISPLAY) (*Howarth et al., 2005*) (0.4 μg each). Following transfection, 5 uM of Biotin was fed to growing cells. 24 hr later the coverslips were transferred to the imaging chamber (ALA MS-512DWPW) in 200 μl physiological salt solution (PSS, ingredients in mM: CaCl2 2, NaCl 151, MgCl2 1, KCl 5, HEPES 10, and Glucose 10, pH 7.3). MNP conjugated with neutravidin-DL550 (0.5 mg/ml) were added (2 μl). After 5 min, the dish was perfused with PSS buffer to remove unbound particles.

## Neuronal culture

Nanoparticles were targeted to the neuronal cell membrane via biotinylated IgM antibody (433110, Invitrogen). Neuronal cultures grown on 12 mm coverslip were loaded in imaging chamber (ALA scientific) holding 200 µl of imaging buffer pH.7.34. Biotinylated antibody (2 µg/ml) was added, incubated for 10 min, before being washed off by perfusing with HEPES buffer pH 7.3. Neutravidin conjugated nanoparticles were then added at 10 µg/ml, and after 5 min the unbound nanoparticles were washed off.

## Molecular cloning and virus packing

Rat TRPV1(VR1) in pcDNA3 was a gift from Dr. Julius (UCSF). The lentivirus vector pLKO.1-EF1a-TRPV1-IRES-DSRed was constructed as follows. A TRPV1 pcr fragment (5' Sal I and 3' Hpa I) was subcloned into pLKO.1-EF1a-IRES-DSRed with a 5' primer (5'-GCG TCG TCG GTC GAC GCC ACC ATG GAA CAA CGG GCT AGC TTA GAC TC-3') and a 3' primer (5'-GCG TCG TCA AGC TTT TAT TTC TCC CCT GGG ACC ATG GAA TC-3'). AAV5-hSyn-TRPV1 was constructed by generating a TRPV1 pcr fragment (5' Sal I and 3' EcoR I) with 5' primer (5'GCG TCG TCG GTC GAC GCC ACC ATG GAA CAA CGG GCT AGC TTA GAC TC-3') and 3' primer (5'-GCG TCG TCG AAT TCT TAT TTC TCC CCT GGG ACC ATG GAA TC-3'). The TRPV1 pcr product was cloned into Sal I and EcoR I of pAAV-hSyn-HA-hM4D(Gi)-mCherry a gift from Bryan Roth (Addgene plasmid #50475). All constructs were sequence verified. The adeno-associated virus, AAV5-hSyn-TRPV1, was generated at the Virus Vector Core at the University of North Carolina at Chapel Hill. The qPCR titer was $1.5 \times 1012$. Packing of the pLKO.1-EF1a-TRPV1-IRES-DSRed plasmid into lentivirus followed established Protocols(*Boyden et al., 2005*).

## In vitro experiments on neurons

### Preparation of neurons

After transfection or nucleofection, the neuronal Cultures were placed in incubator for next 24–48 hr to express the proteins. During this time, 1 µM TTX (Tetrodotoxin) was added to Neuronal culture to minimize endogenous activity. For imaging, the TTX was washed out, and the neurons were placed in Tyrode solution (NaCl 145, CaCl2 2, MgCl2 1, KCl 2.5, HEPES 10, Glucose 20; all in mM). The imaging buffer also contained synaptic blockers DLAP-5 25 µM, NBQX 10 µM, and Gabazine 20 µM, and the osmolality was adjusted to 310–315 mOsmole/L.

### TRPV1 activation by heated buffer

Hippocampal Neuronal cells transiently expressing GCaMP6f and rat TRPV1 were investigated for TRPV1 activation by heat. GCaMP6f acted as marker for nucleofected neurons and as genetically encoded calcium indicator. TRPV1 activation by heat was achieved by perfusion of preheated imaging buffer (HEPES 10 mM, NaCl 140 mM, CaCl2 2 mM, MgCl2 1 mM, KCl 4 mM, Glucose 10 mM). Perfusion buffers were heated using an inline heater (SH-27B, Warner Instruments) with an integrated temperature feedback (TC-324C). Bath temperature was monitored using Neoptix ReFlex optical thermal probe. A custom built microscope enclosure chamber maintained the ambient temperature $33 \pm 1.0°C$. As a measure of temperature-driven TRPV1 activation, transient rise in intracellular $Ca^{2+}$ level was recorded from GCaMP6f fluorescence intensity changes (*Figure 1F*; *Figure 1—figure supplement 2B*; *Figure 3B*).

## Evoking and observing animal behavior

### Animal surgeries for stereotaxic injection of virus or nanoparticles

Male BALB/c mice (3–4 week old, weighing 15–20 g) were obtained from Harlan Laboratories, and housed in the live animal facility (LAF) under 12 hr light-dark cycle in accordance with approved animal protocols from the University at Buffalo SUNY Institutional Animal Care.

Mice were anesthetized by mixture of Ketamine and Xylazine (100 mg of Ketamine and 10 mg of Xylazine per kilogram of body weight i.p.). Burpenorphine (0.1 mg per kg body weight, s.c.) was administered as pre-operative pain medicine. Anesthetized mouse was mounted on Stereotaxic frame (Stoelting) with the help of ear bars and tooth bar. The head was shaved and rubbed with betadine and then ethanol. The skin was retracted and the periosteum was removed at the site. The scalp was opened and hole was drilled in the skull, through which a 33-gauge needle fitted in 5 µl

syringe was inserted into the motor cortex (MC) or striatum1 (St1), or St2, left or right (See supplementary Table 1 for injection coordinates). Using that syringe and an automated syringe pump (World Precision Instruments), 600 nl virus were infused at a rate of 2 nl/s. The injection needle was raised 0.01 mm and kept in place for 5–10 min and slowly removed. The mouse scalp was sutured and anti-inflammatory Carprofen (5 mg/ml) was given for 2 days' post-surgery. All surgical procedures were done under aseptic conditions. Animals were housed for 2 weeks to allow for viral expression before any behavioral experiments were initiated.

### Nanoparticle delivery at targeted site in brain

Equal moles of MNP conjugated with Neutravidin and biotinylated antibody were mixed at 1 mg/ml. The mice were anesthetized using Xylazine and Ketamine, fixed in a stereotactic frame, the periosteum removed, and the pre-drilled hole from the virus injection 2 to 4 weeks earlier was identified. Then, 600 nl of MNP-antibody was injected at the same site as the virus injection 2 weeks back, following exactly the same procedure (see above). The procedure was completed within 45 min and the mice recovered within 1 hr, post anesthesia. Animals were placed in fresh cage and housed for 12 hr before initiating the experiment.

### Recording animal behavior

For each session, the animals were placed in a circular observation area. Sessions were limited to 30 min, including 10 min habituation, and 15–20 min experiment, during which three or four 1-min long AMF applications were given.

A consumer camera (Nikon D810) was used to record video (HD720, 60fps) of the mice before, during and after the magnetic field stimulation. For the 'freezing of gait' response caused by stimulation in the striatum, a different consumer camera (SONY DSC-H50) at 20fps was used. To minimize influence of the light and shadows on the mouse's behavior, the light levels were kept extremely low (camera sensitivity set at ISO 12,800). The videos (12–18 min in length, 1.2–2.4 Gb, uncompressed) were transferred to an iMac27' computer. Red marks were made on the mice with common non-toxic finger paints (Crayola) to aid motion tracking via computer vision.

## Calcium data analysis

Fluorescence microscopy movies were recorded (Micromanager) as TIFF stacks and post processing and ROI (region of interest) mean intensity data were extracted using FIJI (Fiji Is Just ImageJ). Relevant ROIs were cropped from the video files and registration (using the Stack Registration plugin) was done to eliminate any x-y shift in the images over the video frames. Afterwards, an intensity-based threshold operation was done to convert non-cellular dimmer background pixels to NAN (*Figure 3—figure supplement 1A*). A polygonal ROI was then drawn to cover the cell body. The mean intensity data (bit depth) of all the pixels within that ROI was saved with respective frame numbers. The intensity versus time data was then exported to IgorPro, which was used for further analysis.

### Intensity normalization

After importing the mean ROI intensity data as a wave in Igor Pro, constant dark noise values were subtracted from all points in the wave. The dark noise is the mean signal recorded by the camera under no illumination conditions. All experiments were done in the same dark room and the dark noise value deviated little from experiment to experiment and is mostly dependent of the camera exposure times. After, the dark noise cancellation, a further baseline subtraction was done for bleach correction, if substantial bleaching-based decay in baseline was observed. For this, the baseline was fitted to an exponential function and a modified signal was obtained according to $F(t) = F'(t) + \left( F'(0) - F'_{fit}(t) \right)$. Here, $F'(t)$ is the signal values obtained after dark noise cancellation and $F'_{fit}(t))$ is the corresponding exponential fit function. The result, hence obtained was normalized and converted to percentage change in fluorescence. The percentage change in fluorescence intensity was given by:

$$\frac{\Delta F(t)}{F_0} = \frac{F(t) - F_0}{F_0} \times 100$$

where, $F_0$ is the bleach corrected baseline ROI intensity and $F(t)$, the intensity at any time $t$.

### Single spike estimation

The intensity change of GCaMP6f makes it sensitive enough for single action potential detection, but its temporal resolution is not high enough to produce temporally isolated spikes corresponding to each firing in an AP train (*Chen et al., 2013*). To extract action potential events from a GCaMP6f fluorescence signal ($Ca^{2+}$ peaks), we first isolated all peaks from the intensity normalized data corresponding to each neuron with a mean maximum of about 5% (of the baseline). The peaks were then averaged and a mean peak profile was created (*Figure 3—figure supplement 1B,C*). The average peak wave was then interpolated linearly to increase the temporal resolution to 100 Hz.

### AP train reconstruction

A null wave of time length equal to that of the original data, but of 100 Hz frequency (in contrast to 10 Hz, for the original data) was created. The null wave was then converted to a binary wave, with 1 s at the location of estimated AP spikes (*Figure 3—figure supplement 1D,E*; black bars). The estimated single spike wave was then convolved with the binary wave. An overlay of the normalized data and its corresponding convolution-based reconstruction is shown in *Figure 3—figure supplement 1D,E*. A histogram of the residue of the original and the reconstructed waves was fitted to a Gaussian function. The locations of the 1 s in the binary wave were dynamically adjusted to obtain an overall sigma of less than 2%. The binary wave obtained after optimization gave the temporal function of AP events for each recording.

## Behavioral analysis of freely moving mice

### Motion tracking

For all motor behavior analysis, the position of the red neck marker was recorded automatically using the free video analysis and modeling tool TRACKER built by Douglas Brown on the Open Source Physics Java framework. For the active motor responses, every second frame was analyzed, resulting in 30 position measurements per second. The coordinates produced by the tracker were transferred to IgorPro (Wavemetrics) for further analysis and graphing. For motion tracking of the 'freezing of gait' response, the position of each paw was tracked every 1.2 s and the speed of the mouse was calculated as the absolute value of the linear speed averaged over 3 s. For other recordings involving rapid mouse movements, linear speed and an angular speed are reported as an average over 500 ms. To visualize the spatial motion XY-plots of the motion in selected time windows were generated directly from the raw XY coordinates. To visualize the temporal flow, we also plot the XY-coordinates as function of time.

### Video editing for presentation

Videos chosen for publication were edited using VSDC video editor software (Flash-Integro) and Lightworks. Raw videos were compressed using H.264 encoder and downsampled to 15 fps and sped up 3X for fast download and observation. The red marks on the neck were applied to aid motion tracking. Field application periods are indicated on the bottom right, while the MNP injection sites in the brain are indicated on the bottom left.

## Histology

Frozen coronal Rostral to caudal brain sections were prepared following standard protocols. The 75-μm-thick slices were stained for subsequent laser confocal imaging.

Soma and dendrites of neurons were marked using a rabbit anti-microtubules-associated- protein (MAP2) antibody (ab32454, Abcam)(24 hr, 1:500), in conjunction with anti-rabbit IgG Alexa 488 (ab150077, Abcam)(1 hr).

MNPs coated with PMA were conjugated with Neutravidin, tagged with ATTO 488. To boost the fluorescence signal, some brain slices were incubated with Biotin-PEG, tagged with Alexa 647 dye (24 hr, 1:500).

AAV5 or lenti-virus was used for TRPV1 channel delivery to the brain. The lenti-virus delivered also a nuclear marker marker (NLS-mCherry). As the AAV5 did not contain any marker, the AAV slices were incubated with anti-TRPV1 antibody (ab31895, Abcam), which we directly conjugated with Alexa 647.

## Acknowledgement

We thank Wolfgang J Parak (University at Marburg, Germany) for a gift of MNPs, Mathew Paul (University at Buffalo) for lending us his micro-pump, Jason Myers (University at Buffalo) for molecular cloning and preparation of cultured hippocampal neurons, and Sara Parker (University at Buffalo) for the histology. TPRV1 was a gift from David Julius (UCSF), pGP-CMV-GCaMP6f a gift from Douglas Kim (Addgene plasmid # 40755), and GL-GPI-mCherry from Gerald Baron (US Dept. of Health and Human Services). This work was supported by NIH grants was supported by HSFP project grant RGP0052/2012 and NIH grant R01MH094730.

## Additional information

### Funding

| Funder | Grant reference number | Author |
| --- | --- | --- |
| National Institute of Mental Health | 1R01MH094730 | Arnd Pralle |
| Human Frontier Science Program | RGP0052/2012 | Arnd Pralle |
| National Institute of Mental Health | 1R01MH111872 | Arnd Pralle |

The funders had no role in study design, data collection and interpretation, or the decision to submit the work for publication.

### Author contributions

Rahul Munshi, Formal analysis, Validation, Investigation, Visualization, Methodology, Writing—review and editing; Shahnaz M Qadri, Data curation, Investigation, Methodology, Writing—original draft, Writing—review and editing; Qian Zhang, Idoia Castellanos Rubio, Pablo del Pino, Resources; Arnd Pralle, Conceptualization, Data curation, Supervision, Funding acquisition, Validation, Methodology, Project administration, Writing—review and editing

### Author ORCIDs

Rahul Munshi http://orcid.org/0000-0002-8900-2816
Shahnaz M Qadri http://orcid.org/0000-0001-8818-4753
Pablo del Pino http://orcid.org/0000-0003-1318-6839
Arnd Pralle http://orcid.org/0000-0002-6079-109X

### Ethics

Animal experimentation: This study was performed in strict accordance with the recommendations in the Guide for the Care and Use of Laboratory Animals of the National Institutes of Health. All of the animals were handled according to approved institutional animal care and use committee (IACUC) protocols (PHY01051Y and PHY02103N) of the University at Buffalo. The protocol was approved by the Committee on the Ethics of Animal Experiments of the University at Buffalo. All surgery was performed and Ketamine anesthesia, and every effort was made to minimize suffering.

### Decision letter and Author response

Decision letter https://doi.org/10.7554/eLife.27069.042
Author response https://doi.org/10.7554/eLife.27069.043

## Additional files

### Supplementary files
• Transparent reporting form
DOI: https://doi.org/10.7554/eLife.27069.040

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
