## [Decision Letter]

Thank you for submitting your article "Magneto-thermal genetic deep brain stimulation of motor behaviors in awake, freely moving mice" for consideration by *eLife*. Your article has been reviewed by three peer reviewers, and the evaluation has been overseen by a Reviewing Editor and Richard Aldrich as the Senior Editor. One of the three reviewers, Jinwoo Cheon, has agreed to reveal his identity.

The reviewers have discussed the reviews with one another and the Reviewing Editor has drafted this decision to help you prepare a revised submission.

Summary:

This manuscript reports characterization of genetically encoded magnetothermal brain stimulation and applications to behaving mice. Immunochemical tethering of polymer coated magnetic nanoparticles to neurons is coupled to pan-neuronal Trpv1 expression, which renders neurons sensitive to heating in an alternating magnetic field (AMF). The authors nicely characterize the relationship between magnetic nanoparticle number or field strength and heating, as well as production of calcium transients in neurons. The authors go on to transduce three different brain areas with Trpv1 and demonstrate magnetic-field dependent alterations in movement that are consistent with other neuronal activation studies using different tools. The magnetogenetic system described here appears to be a promising tool for in vivo applications, but additional practical information about repeatability, cell type specificity, and cell health are required before it is suitable for publication in *eLife*.

Additional behavioral and histological analyses are required, potentially requiring additional experiments.

Major revisions:

*Behavior*

1) The authors show a small number of example perturbation trials (1-4). A larger number of trials (~10) in a single daily session should be shown. Data should be reported for each within session trial to demonstrate the consistency of the response, beyond the anecdotal examples provided.

2) The effectiveness across multiple sessions (different days) should be demonstrated. What is the timepoint after nanoparticle delivery after which responsiveness declines due to clearance of magnetic particles or antibody degradation? Can an additional injection of nanoparticles renew sensitivity at this point?

3) The authors make a compelling case that the membrane targeting is more efficient than soluble MNP in culture systems. However, it is not clear how this translates to the in-vivo situation. It could be that normal MNPs (without AB treatment) have the same or even better efficiency in-vivo. A comparison of AB and non-AB treated MNP, for the behavioral effects, would be strengthen this point.

4) Ex vivo experiments use 5 s AMF stimulation but for in vivo experiments AMF stimulation is longer (the precise stimulus durations seems to be absent). The effect of these longer AMF stimuli should also be characterized in cultured cells at similar field strength as in the brain.

*Histology*

1) Histological analysis should show that AB treatment effectively enhances membrane targeting in-vivo in a cell-type specific manner.

2) It should be examined that number of nanoparticles per cells and expression level of ion channels are comparable for in vivo and in vitro conditions. It is also important to show that injected TRPV1 virus produces expression in mouse brains.

3) A major worry is inflammation and heating. Histology showing healthy, MNP-covered neurons after stimulation would make this more convincing.

4) Another major concern is local heating in the brain. In Figure 1, it seems that the nanoparticles form aggregates (big red spots). How much stable nanoparticles are in physiological buffer? How clumped are the particles in the brain? Close to clumps the temperature could increase substantially and cause damage.

---

## [Author Response]

Major revisions:

Behavior

1) The authors show a small number of example perturbation trials (1-4). A larger number of trials (~10) in a single daily session should be shown. Data should be reported for each within session trial to demonstrate the consistency of the response, beyond the anecdotal examples provided.

To avoid stressing the animals, our approved animal behavior protocol had limited each behavior session to 30 minutes. This included ten minutes habituation, leaving at most 20 minutes for experiments. To avoid any local heat accumulation, each 1 minute stimulation trial was followed by at least 3 minutes break. Therefore, each session contained three to four trials. However, for several animals, we have performed two sessions per day and continued this pattern for multiple days, achieving more than ten trials.

We now include an overlay of many behavioral responses from multiple trials and animals in Figure 4—figure supplement 1 and in Figure 5. We also should the tracking results for multiple trials across multiple sessions and day for one animal in Figure 4—figure supplement 4 and Figure 5—figure supplement 4.

2) The effectiveness across multiple sessions (different days) should be demonstrated. What is the timepoint after nanoparticle delivery after which responsiveness declines due to clearance of magnetic particles or antibody degradation? Can an additional injection of nanoparticles renew sensitivity at this point?

We have now included tracking data for two animals (one motor cortex, one striatum) observed for several sessions over multiple days in Figure 4—figure supplement 4 and Figure 5—figure supplement 4. Animals consistently respond for more than two days (54 hours). At the 54 hours time point, one field stimulation fails to elicit a response which may hint some clearing of the nanoparticles from the location where TRPV1 is expressed, but further studies will need to confirm this.

As the histology (see below) shows no cellular damage after twenty AMF stimulations, it may very well be possible to renew sensitivity by an additional injection of nanoparticles. Unfortunately, confirming this requires more time and animals than we had to answer these comments.

3) The authors make a compelling case that the membrane targeting is more efficient than soluble MNP in culture systems. However, it is not clear how this translates to the in-vivo situation. It could be that normal MNPs (without AB treatment) have the same or even better efficiency in-vivo. A comparison of AB and non-AB treated MNP, for the behavioral effects, would be strengthen this point.

We now include behavioral data of three animals expressing TRPV1 and injected with the same amount and type of nanoparticles, but without the antibody. These animals do not respond to the application of the AMF (Figure 5 and Figure 4—figure supplement 3). Further evidence for the effectiveness of the AB targeting is seen in the now added histology data (see below).

4) Ex vivo experiments use 5 s AMF stimulation but for in vivo experiments AMF stimulation is longer (the precise stimulus durations seems to be absent). The effect of these longer AMF stimuli should also be characterized in cultured cells at similar field strength as in the brain.

The in vivo experiments used one-minute AMF stimulation (which was mentioned in the Results and Discussion section but omitted in the Materials and methods. We have added it there now as well). In ex vivo experiments, the AMF was strong enough that the cells responded within 2-3 s, so that a 5-s AMF stimulation was sufficient. For the in vivo experiments, the field strength was lower and a response was observed only after 15-20s. We share the reviewer’s concern that it is necessary to confirm that the heat remains sufficiently localized when the heating occurs at a slower rate. We now include a series local temperature measurements at the membrane of cells decorated with nanoparticles heated by AMF of two different magnitudes and durations. During the 5 s AMF used for the ex vivo data, the local temperature rises nearly linearly by 2 degrees, showing little evidence of the cooling to the environment. After the field is turned off the temperature cools off quickly. During the 30 s applications of the lower AMF strength used in vivo, the local temperature rises more slowly and the cooling to the environment is apparent as the temperature rise slows. Still, the local temperature does increases by 2 degrees, and cools to baseline after removing the AMF. The data is included as Figure 2—figure supplement 1.

Histology

1) Histological analysis should show that AB treatment effectively enhances membrane targeting in-vivo in a cell-type specific manner.

We have performed extensive histological analysis on mice injected with MNP and AB as well as some without AB. On the macroscale, it is apparent that the presence of the AB reduces how far the MNP diffuse into the brain, indicating efficient binding to cells (Figure 5—figure supplement 5). Without AB, the MNP diffuse further into the brain.

Stacks of higher resolution confocal images labeled for the MNP and the neuronal marker MAP2 demonstrate specific binding of the MNP to the membrane of neurons, on the soma and along processes of these neurons (Figure 5—figure supplement 6, including videos of z-sections). Without AB treatment, we did not find any examples with MNPs on the membrane of neurons.

We also include histological images demonstrating overlap of TRVP1 expression in neurons with MNP labeling via AB binding (Figure 4—figure supplement 2).

2) It should be examined that number of nanoparticles per cells and expression level of ion channels are comparable for in vivo and in vitro conditions. It is also important to show that injected TRPV1 virus produces expression in mouse brains.

We now include confocal images of brain sections showing the expression of TRPV1 in the motor cortex (Figure 4—figure supplement 2) and in the striatum (Figure 5—figure supplement 3), both using AVV5 and lenti-virus to deliver the gene.

Currently, we confirm TRPV1 expression and nanoparticle targeting using fluorescence imaging. This permits only a rough comparison of the actual expression levels in vivo and in vitro, but they appear comparable. A proper quantification is beyond the scope of this manuscript.

3) A major worry is inflammation and heating. Histology showing healthy, MNP-covered neurons after stimulation would make this more convincing.

We include confocal images of post-experiment brain sections, after 20 AMF applications, showing intact neurons labeled with MNP on their membrane (Figure 5—figure supplement 6). These demonstrate that the heating does not damage the neurons. This is also supported by the ability to repeatedly magnetothermally evoke behavior, and that animals show no abnormal behavior after AMF application.

While we do not observe wide-spread inflammation, the histology does show some local inflammation at the injection wound and 24 hours post injection some MNP can be found in dendritic immune cells in lymph adjacent to blood vessels (faint dots far from injection in Figure 5—figure supplement 1,Figure 5—figure supplement 2).

4) Another major concern is local heating in the brain. In Figure 1, it seems that the nanoparticles form aggregates (big red spots). How much stable nanoparticles are in physiological buffer? How clumped are the particles in the brain? Close to clumps the temperature could increase substantially and cause damage.

The histology sections show that the MNP are stable in the physiological milieu of the brain with minimal clustering. Some minor clustering outside of cells does occur. Some of those clusters do not heat because the MNP magnetically interact and are not in resonance with the AMF anymore. Other clusters may heat more efficiently. However, using histology we found no evidence for damage but cannot completely exclude very local damage affecting individual cells. Any damage would be very local as the temperature drops around them very quickly reaching background levels within 500 nanometers.